# Generation from Noisy Examples

**Ananth Raman** [* 1]  **Vinod Raman** [* 2]

## Abstract

We continue to study the learning-theoretic foundations of generation by extending the results from Kleinberg & Mullainathan (2024) and Li et al. (2024) to account for *noisy* example streams. In the *noiseless* setting of Kleinberg & Mullainathan (2024) and Li et al. (2024), an adversary picks a hypothesis from a binary hypothesis class and provides a generator with a sequence of its positive examples. The goal of the generator is to eventually output new, unseen positive examples. In the *noisy* setting, an adversary still picks a hypothesis and a sequence of its positive examples. But, before presenting the stream to the generator, the adversary *inserts* a finite number of negative examples. Unaware of which examples are noisy, the goal of the generator is to still eventually output new, unseen positive examples. In this paper, we provide necessary and sufficient conditions for when a binary hypothesis class can be noisily generatable. We provide such conditions with respect to various constraints on the number of distinct examples that need to be seen before perfect generation of positive examples. Interestingly, for finite and countable classes we show that generatability is largely unaffected by the presence of a finite number of noisy examples.

## 1. Introduction

Generation is an important paradigm in machine learning with promising applications to natural language processing (Wolf et al., 2020), computer vision (Khan et al., 2022), and computational chemistry (Vanhaelen et al., 2020). In contrast to its practical applications, a strong theoretical foundation of generation is largely missing from literature. Recently, Kleinberg & Mullainathan (2024), inspired by the seminal work of E Mark Gold on language identification

in the limit (Gold, 1967), introduced a theoretical model of language generation called "Language Generation in the Limit." In this model, there is a countable set $U$ of strings and a countable language family $C = \{L_1, L_2, \dots\}$, where $L_i \subseteq U$ for all $i \in \mathbb{N}$. An adversary picks a language $K \in C$ and begins to enumerate the strings one by one to the player in rounds $t = 1, 2, \dots$. After observing the string $w_t$ in round $t \in \mathbb{N}$, the player guesses a string $\hat{w}_t \in U$ in the hope that $\hat{w}_t \in K \setminus \{w_1, \dots, w_t\}$. The player has generated from $K$ in the limit, if there exists a finite time step $t \in \mathbb{N}$ such that for all $s \geq t$, we have that $\hat{w}_s \in K \setminus \{w_1, \dots, w_s\}$. The class $C$ is generatable in the limit, if the player can generate from all $K \in C$. Unlike language identification in the limit, Kleinberg & Mullainathan (2024) prove that generation in the limit is possible for *every* countable language family $C$.

Kleinberg & Mullainathan (2024)'s positive result has spawned a surge of new work, extending the results of Kleinberg & Mullainathan (2024) in various ways (Kalavasis et al., 2024b; Li et al., 2024; Charikar & Pabbaraju, 2024; Kalavasis et al., 2024a). Of particular interest to us is the work by Li et al. (2024). They frame the results of Kleinberg & Mullainathan (2024) through a learning-theoretic lens by taking the language family $C$ to be a binary hypothesis class $\mathcal{H} \subseteq \{0, 1\}^{\mathcal{X}}$ defined over some countable instance space $\mathcal{X}$. By doing so, Li et al. (2024) identify stronger notions of generatability, termed "uniform" [1] and "non-uniform" generatability, and provide characterizations of which classes are uniformly and non-uniformly generatable in terms of new combinatorial dimensions.

A commonality of Kleinberg & Mullainathan (2024) and its follow-up work is the assumption that the adversary presents to the player a "noiseless" stream of examples – one where every example/string must be contained in the hypothesis/language chosen by the adversary. In practice, such an assumption is unrealistic, as one would still like to generate well even if a few examples in the dataset are imperfect. For example, Large Language Models (LLM) are often trained using the potentially hallucinatory outputs of other LLMs (Burns et al., 2023; Briesch et al., 2023). More generally, one might want a generative model to be robust

---

[*]Equal contribution   [1]Bridgewater-Raritan Regional High School [2]University of Michigan, Ann Arbor. Correspondence to: Vinod Raman <vkraman@umich.edu>.

*Proceedings of the 42nd International Conference on Machine Learning*, Vancouver, Canada. PMLR 267, 2025. Copyright 2025 by the author(s).

---

[1]Uniform generatability was also informally considered by Kleinberg & Mullainathan (2024)

to data contamination/poisoning attacks (Shang et al., 2018; Zhang et al., 2023; Jiang et al., 2023).

Motivated by these concerns, we study generation, in the framework posed by Kleinberg & Mullainathan (2024) and Li et al. (2024), under *noisy* example streams. In particular, we focus on a simple, but natural noising process: after selecting a positive example stream, the adversary is allowed to *insert* a finite number of negative examples in any way it likes, *unbeknownst* to the player. Such a noising process, as well as others, have been extensively studied in the context "language identification in the limit" or "inductive inference" (Schäfer, 1985; Fulk & Jain, 1989; Baliga et al., 1992; Jain, 1994; Stephan, 1997; Case et al., 1997; Lange & Grieser, 2002; Mukouchi & Sato, 2003; Tantini et al., 2006). In this paper, we extend this study to *generation*. To that end, our main contributions are summarized below.

(1) We extend the notions of uniform generatability, non-uniform generatability, and generatability in the limit from Kleinberg & Mullainathan (2024) and Li et al. (2024) to account for finite, noisy example streams. We call these new settings "uniform noise-dependent generatability," "non-uniform noise-dependent generatability," and "noisy generatability in the limit" respectively.

(2) We provide a complete characterization of which classes are uniformly noise-dependent generatable in terms of a new scale-sensitive dimension we call the Noisy Closure dimension.

*Theorem* (Informal). A class $\mathcal{H} \subseteq \{0,1\}^{\mathcal{X}}$ is *uniformly noise-dependent generatable* if and only if $\mathrm{NC}_n(\mathcal{H}) < \infty$ for every $n \in \mathbb{N}$, where $\mathrm{NC}_n(\mathcal{H})$ is the *Noisy Closure dimension* of $\mathcal{H}$ at noise-level $n$.

We show that uniform noise-dependent generation is strictly harder than noiseless uniform generation. In fact, we construct a countably infinite class which is trivially uniformly generatable when there is no noise, but not uniformly noise-dependent generatable even when the adversary is only allowed to perturb a single example. Despite this hardness, we show that all finite classes are still uniformly noise-dependent generatable.

(3) We provide a sufficient condition for which classes are non-uniformly noise-dependent generatable.

*Lemma* (Informal). A class $\mathcal{H} \subseteq \{0,1\}^{\mathcal{X}}$ is *non-uniformly noise-dependent generatable* if there exists a non-decreasing sequence of classes $\mathcal{H}_1 \subseteq \mathcal{H}_2 \subseteq \ldots$ such that $\mathcal{H} = \bigcup_{i=1}^{\infty} \mathcal{H}_i$ and $\mathrm{NC}_i(\mathcal{H}_i) < \infty$ for all $i \in \mathbb{N}$.

Using the result that all finite classes are uniformly noise-dependent generatable, we prove as a corollary

that all countable classes are non-uniformly noise-dependent generatable. Since non-uniform noise-dependent generation implies noisy generation in the limit, this corollary also implies that all countable classes are noisily generatable in the limit. Although not matching, we also provide a necessary condition for non-uniform noise-dependent generation in terms of the Noisy Closure dimension.

*Lemma* (Informal). A class $\mathcal{H} \subseteq \{0,1\}^{\mathcal{X}}$ is *non-uniformly noise-dependent generatable* only if for every $n \in \mathbb{N}$, there exists a non-decreasing sequence of classes $\mathcal{H}_1 \subseteq \mathcal{H}_2 \subseteq \ldots$ such that $\mathcal{H} = \bigcup_{i=1}^{\infty} \mathcal{H}_i$ and $\mathrm{NC}_n(\mathcal{H}_i) < \infty$ for all $i \in \mathbb{N}$.

We leave the complete characterization of non-uniform noise-dependent generatability as an open question.

(4) We provide two different sufficiency conditions for noisy generatability in the limit. The first shows that (noiseless) non-uniform generatability is sufficient for noisy generatability in the limit.

*Theorem* (Informal). If a class $\mathcal{H} \subseteq \{0,1\}^{\mathcal{X}}$ is (noiseless) non-uniformly generatable, then it is noisily generatable in the limit.

Since Li et al. (2024) prove that all countable classes are (noiseless) non-uniform generatable, this result also shows that all countable classes are noisily generatable in the limit. In addition, we also give a sufficiency condition in terms of an even stronger notion of noisy generatability called "uniform noise-independent generatability."

*Theorem* (Informal). If there exists a finite sequence of uniformly noise-independent generatable classes $\mathcal{H}_1, \mathcal{H}_2, \ldots, \mathcal{H}_k$ such that $\mathcal{H} = \bigcup_{i=1}^{k} \mathcal{H}_i$, then $\mathcal{H}$ is noisily generatable in the limit.

One can think of this latter sufficiency condition as the analog of Theorem 3.10 in Li et al. (2024) which shows that the ability to write a class as the finite union of (noiseless) uniformly generatable classes is sufficient for (noiseless) generatability in the limit.

While our theorem statements look similar to that of Li et al. (2024), our proof techniques are different due to the fact that the Noisy Closure dimension is scale-sensitive. This difference manifests even when characterizing noisy uniform generation. In particular, our uniform generator effectively requires combining different generators for each noise-level whereas the noiseless uniform generator from Li et al. (2024) does not.

## 1.1. Related Work

**Language Identification in the Limit.** In his seminal 1967 paper, E Mark Gold introduced the model of "Language

Identification in the Limit (Gold, 1967)." In this model, there is a countable set $U$ of strings and a countable language family $C = \{L_1, L_2, \dots\}$, where $L_i \subseteq U$ for all $i \in \mathbb{N}$. An adversary picks a language $K \in C$, and begins to enumerate the strings in $K$ one by the one to the player in rounds $t = 1, 2, \dots$. After observing the string $w_t$ in round $t \in \mathbb{N}$, the player guesses an index $i_t \in \mathbb{N}$ with the hope that $L_{i_t} = K$. The player has identified $K$ in the limit, if there exists a finite time step $t \in \mathbb{N}$ such that for all $s \geq t$, we have that $L_{i_s} = K$. The class $C$ is identifiable in the limit, if the player can identify all $K \in C$. Gold showed that while all finite language families are identifiable in the limit, there are simple countable language families which are not. In a follow-up work, Angluin (1979; 1980) provides a precise characterization of language families $C$ for which language identification in the limit is possible. The results by Gold and Angluin emphasized the impossibility of language identification in the limit by ruling out the vast majority of language families. Since Gold's seminal work on language identification in the limit, there has been extensive follow-up work on this model and we refer the reader to the excellent survey by Lange et al. (2008).

**Language Identification from Noisy Examples.** In addition to the work by Angluin (1979; 1980), a different line of follow-up work focused on extending Gold's model to account for noisy example streams. There have been several proposed noise models, and we review a few of them here. The most standard noise model for language identification in the limit allows the adversary to first pick an enumeration of its chosen language $K \in L$ and then *insert* a *finite* number of strings that do not belong to $K$ (Baliga et al., 1992; Fulk & Jain, 1989; Schäfer, 1985; Case et al., 1997; Stephan, 1997; Lange & Grieser, 2002). We use the term *insert* to emphasize that every $w \in K$ must still be present in the sequence chosen by the adversary. Such noisy streams are often referred to as *noisy texts*. Jain (1994) go beyond the finite nature of the noise by considering sequences of strings with an infinite number of noisy incorrect strings, where the amount of noise is measured using certain density notions. In a different direction, Mukouchi & Sato (2003) and Tantini et al. (2006) model noise at the string-level by defining a distance metric on the space of all strings and allowing the adversaries to insert and delete strings in the original enumeration of $K$ which are at most some distance away from some true string in $K$.

**Language Generation in the Limit.** Inspired by large language models and recent interest in generative machine learning, Kleinberg & Mullainathan (2024) study the problem of language *generation* in the limit. In this problem, the adversary also picks a language $K \in C$, and begins to enumerate the strings one by one to the player in rounds $t = 1, 2, \dots$. However, now, after observing the string $w_t$ in round $t \in \mathbb{N}$, the player guesses a string $\hat{w}_t \in U$ with the

hope that $\hat{w}_t \in K \setminus \{w_1, \dots, w_t\}$. The player has generated from $K$ in the limit, if there exists a finite time step $t \in \mathbb{N}$ such that for all $s \geq t$, we have that $\hat{w}_s \in K \setminus \{w_1, \dots, w_s\}$. Kleinberg & Mullainathan (2024) prove a strikingly different result than Gold and Angluin – generation in the limit is possible for *every* countable language family $C$. This positive result has spurred a number of follow-up works, which we briefly review below.

Kalavasis et al. (2024b) study generation in the stochastic setting, where the positive examples revealed to the generator are sampled i.i.d. from some unknown distribution. In this model, they study the trade-offs between generating with breadth and generating with consistency and show that in general, achieving both is impossible, resolving an open question posed by Kleinberg & Mullainathan (2024) for a large family of language models. More recently, Charikar & Pabbaraju (2024) and Kalavasis et al. (2024a) further formalize this tension between consistency and breadth by defining various notions of breadth and providing complete characterizations of which language families are generatable in the limit with breadth.

In a different direction, and most closely related to our work, Li et al. (2024) reinterpret the results of Kleinberg & Mullainathan (2024) through a binary hypothesis class $\mathcal{H} \subseteq \{0, 1\}^{\mathcal{X}}$ defined over a countable example space $\mathcal{X}$. By doing so, Li et al. (2024) extend the results of Kleinberg & Mullainathan (2024) beyond language generation while also formalizing two stronger settings of generation they term "uniform" and "non-uniform" generation. Unlike Kleinberg & Mullainathan (2024) and Kalavasis et al. (2024a) who focus on computable learners and countable language families, Li et al. (2024) place no such restrictions and provide complete, information-theoretic characterizations of which hypothesis classes are uniformly and non-uniformly generatable. Li et al. (2024) leave the characterization of generatability in the limit as an open question (see Section 4).

## 2. Preliminaries

Let $\mathcal{X}$ denote a *countable* example space and $\mathcal{H} \subseteq \{0, 1\}^{\mathcal{X}}$ denote a binary hypothesis class. Let $\mathcal{X}^\star$ denote the set of all finite subsets of $\mathcal{X}$. Let $[N] := \{1, \dots, N\}$ and abbreviate a finite sequence $x_1, \dots, x_n$ as $x_{1:n}$. For any $h \in \mathcal{H}$, an *enumeration* of $\mathrm{supp}(h)$ is any infinite sequence $x_1, x_2, \dots$ such that $\bigcup_{i \in \mathbb{N}} \{x_i\} = \mathrm{supp}(h)$. In other words, for every $x \in \mathrm{supp}(h)$, there exists an $i \in \mathbb{N}$ such that $x_i = x$ and for every $i \in \mathbb{N}$, we have that $x_i \in \mathrm{supp}(h)$. For any $h \in \mathcal{H}$, a *noisy* enumeration of $\mathrm{supp}(h)$ is any infinite sequence $x_1, x_2, \dots$ such that for every $x \in \mathrm{supp}(h)$, there exists an $i \in \mathbb{N}$ such that $x_i = x$ and $\sum_{t=1}^{\infty} \mathbb{1}\{x_t \notin \mathrm{supp}(h)\} < \infty$. For a finite sequence of examples $x_1, \dots, x_d$ and $n \in \mathbb{N} \cup \{0\}$, define $\mathcal{H}(x_{1:d}; n) := \{h \in \mathcal{H} : |\{x_{1:d}\} \cap \mathrm{supp}(h)| \geq$

$d - n$}. For any class $\mathcal{H}$, define its induced closure operator at noise-level $n$ as $\langle \cdot \rangle_{\mathcal{H},n}$ such that

$$\langle x_{1:d} \rangle_{\mathcal{H},n} := \begin{cases} \bigcap_{h \in \mathcal{H}(x_{1:d};n)} \text{supp}(h), & \text{if } |\mathcal{H}(x_{1:d};n)| \geq 1 \\ \bot, & \text{if } |\mathcal{H}(x_{1:d};n)| = 0 \end{cases}.$$

Note that $\langle \cdot \rangle_{\mathcal{H},0}$ is exactly the closure operator from Section 2.1 of Li et al. (2024). We will make the following assumption about hypothesis classes.

**Assumption 2.1** (Uniformly Unbounded Support (UUS) (Li et al., 2024)). A hypothesis class $\mathcal{H} \subseteq \{0,1\}^{\mathcal{X}}$ satisfies the *Uniformly Unbounded Support (UUS) property* if $|\text{supp}(h)| = \infty$ for every $h \in \mathcal{H}$.

*Remark* 2.2. The UUS assumption is mainly needed for bookkeeping purposes to prevent the adversary from exhausting all positive examples and thus making the task of generating unseen positive examples impossible. This is consistent with the assumption that each language is countably infinite in size in the work of Kleinberg & Mullainathan (2024).

## 2.1. Generatability

In this paper, we adopt the learning-theoretic framework for generation introduced by Li et al. (2024). To that end, we define a generator as a deterministic map from a finite sequence of examples to a new example.

**Definition 2.3** (Generator). A generator is a map $\mathcal{G} : \mathcal{X}^{\star} \to \mathcal{X}$ that takes a finite sequence of examples $x_1, x_2, \ldots$ and outputs a new example $x$.

### 2.1.1. NOISELESS GENERATION

In the noiseless setting, an adversary plays a game against a generator $\mathcal{G}$. Before the game begins, the adversary picks a hypothesis $h \in \mathcal{H}$ and a sequence of examples $x_1, x_2, \ldots$ such that $\{x_1, x_2, \ldots\} \subseteq \text{supp}(h)$. The adversary reveals the examples as a stream to $\mathcal{G}$ one at a time, and the goal of the generator is to *eventually* output new, unseen positive examples $\hat{x}_t \in \text{supp}(h) \setminus \{x_1, \ldots, x_t\}$.

Depending on how one quantifies "eventually," one can get various notions of noiseless generatability. For example, if one requires the generator to perfectly generate new unseen examples after observing $d \in \mathbb{N}$ examples regardless of the hypothesis or stream chosen by the adversary, this is called "uniform generation." If the number of positive examples required before perfect generation can depend on the hypothesis chosen by the adversary, but not the stream, then this is "non-uniform generation." Finally, if the number of positive examples before perfect generation can depend on both the hypothesis and the stream selected by the adversary, this is called "generation in the limit." We refer the reader to Appendix A for complete definitions and Appendix B for a summary of results.

### 2.1.2. NOISY GENERATION

In the noisy model, we consider the following game. Like before, the adversary picks a hypothesis $h \in \mathcal{H}$, and a sequence of positive examples $z_1, z_2, \cdots \in \text{supp}(h)$. But now, the adversary picks a noise-level $n^{\star} \in \mathbb{N}$, and *inserts* at most $n^{\star}$ negative examples in $z_1, z_2, \ldots$ to obtain a *noisy* stream $x_1, x_2, \ldots$. The adversary then presents the examples in the noisy stream to generator $\mathcal{G}$ one at a time. Without knowledge of the noise-level or the location of the negative examples, the goal of the generator is to *eventually* output new, positive examples $\hat{x}_t \in \text{supp}(h) \setminus \{x_1, \ldots, x_t\}$.

Like the noiseless case, there are several definitions for noisy generatability based on how one quantifies "eventually." The most extreme case is what we term "uniform noise-independent generatability."

**Definition 2.4** (Uniform Noise-Independent Generatability). Let $\mathcal{H} \subseteq \{0,1\}^{\mathcal{X}}$ be any hypothesis class satisfying the UUS property. Then, $\mathcal{H}$ is *uniformly noise-independent generatable*, if there exists a generator $\mathcal{G}$ and $d^{\star} \in \mathbb{N}$, such that for every $h \in \mathcal{H}$ and any sequence $x_1, x_2, \ldots$ with $\sum_{t=1}^{\infty} \mathbb{1}\{x_t \notin \text{supp}(h)\} < \infty$, if there exists $t^{\star} \in \mathbb{N}$ such that $|\{x_1, \ldots, x_{t^{\star}}\}| = d^{\star}$, then $\mathcal{G}(x_{1:s}) \in \text{supp}(h) \setminus \{x_1, \ldots, x_s\}$ for all $s \geq t^{\star}$.

Here, the generator must perfectly generate after seeing $d^{\star} \in \mathbb{N}$ examples, where $d^{\star}$ can only depend on the class $\mathcal{H}$ itself, and thus is uniform over the noise-level $n^{\star}$, the hypothesis $h \in \mathcal{H}$, and the stream $x_1, x_2, \ldots$ selected by the adversary. Unsurprisingly, as we show in Section 3.1, uniform noise-independent generatability is impossible even for simple classes with just two hypotheses. In Appendix C, we show this is also the case if we measure sample complexity in terms of just the number of distinct *positive* examples (as opposed to all examples). These results motivate weakening the definition by allowing $d^{\star}$ to depend on the noise-level. We call this setting "Uniform Noise-dependent Generatability."

**Definition 2.5** (Uniform Noise-dependent Generatability). Let $\mathcal{H} \subseteq \{0,1\}^{\mathcal{X}}$ be any hypothesis class satisfying the UUS property. Then, $\mathcal{H}$ is *uniformly noise-dependent generatable*, if there exists a generator $\mathcal{G}$ such that for every noise-level $n^{\star} \in \mathbb{N}$, there exists a $d^{\star} \in \mathbb{N}$, such that for every $h \in \mathcal{H}$ and any sequence $x_1, x_2, \ldots$ with $\sum_{t=1}^{\infty} \mathbb{1}\{x_t \notin \text{supp}(h)\} \leq n^{\star}$, if there exists $t^{\star} \in \mathbb{N}$ such that $|\{x_1, \ldots, x_{t^{\star}}\}| = d^{\star}$, then $\mathcal{G}(x_{1:s}) \in \text{supp}(h) \setminus \{x_1, \ldots, x_s\}$ for all $s \geq t^{\star}$.

Uniform noise-dependent generatability does not suffer from the same hardness of uniform noise-independent generatability. In fact, in Section 3.2, we show that all finite classes are uniformly noise-dependent generatable. Nevertheless, we can continue weakening the notion of noisy generatability, by allowing $d^{\star}$ to also depend on the hypoth-

esis chosen by the adversary.

**Definition 2.6** (Non-uniform Noise-dependent Generatability). Let $\mathcal{H} \subseteq \{0,1\}^{\mathcal{X}}$ be any hypothesis class satisfying the UUS property. Then, $\mathcal{H}$ is *non-uniformly noise-dependent generatable*, if there exists a generator $\mathcal{G}$, such that for every noise-level $n^{\star} \in \mathbb{N}$ and any $h \in \mathcal{H}$ there exists $d^{\star} \in \mathbb{N}$ such that for any sequence $x_1, x_2, \ldots$ with $\sum_{t=1}^{\infty} \mathbb{1}\{x_t \notin \mathrm{supp}(h)\} \leq n^{\star}$, if there exists $t^{\star} \in \mathbb{N}$ such that $|\{x_1, \ldots, x_{t^{\star}}\}| = d^{\star}$, then $\mathcal{G}(x_{1:s}) \in \mathrm{supp}(h) \setminus \{x_1, \ldots, x_s\}$ for all $s \geq t^{\star}$.

The term "non-uniform" here is used to refer to the fact that the number of unique examples needed by the generator before perfect generation can depend on the selected $h \in \mathcal{H}$, and hence it is "non-uniform" over the hypothesis class, unlike the previous two definitions. Finally, the weakest form of noisy generation allows $d$ to depend on the noise-level, hypothesis, and stream selected by the adversary.

**Definition 2.7** (Noisy Generatability in the Limit). Let $\mathcal{H} \subseteq \{0,1\}^{\mathcal{X}}$ be any hypothesis class satisfying the UUS property. Then, $\mathcal{H}$ is *noisily generatable in the limit*, if there exists a generator $\mathcal{G}$, such that for every $h \in \mathcal{H}$ and any *noisy* enumeration $x_1, x_2, \ldots$ of $\mathrm{supp}(h)$, there exists $t^{\star} \in \mathbb{N}$ such that $\mathcal{G}(x_{1:s}) \in \mathrm{supp}(h) \setminus \{x_1, \ldots, x_s\}$ for all $s \geq t^{\star}$.

Note that in Definition 2.7, we require the noisy stream picked by the adversary to still contain every example in the support of the selected hypothesis. This is consistent with the model of "noisy texts" from the literature in language identification where one allows the adversary to *insert* noisy examples in the stream, as opposed to *replacing* positive examples (Stephan, 1997).

*Remark* 2.8. The astute reader might notice that there is actually a fifth setting of noisy generatability that we did not cover. In this fifth setting, which we will call "Non-uniform Noise-independent Generatability," the number of positive examples needed before perfect generation can depend on the hypothesis chosen by the adversary, but must still be uniform over the noise-level and the noisy example stream chosen by the adversary. However, as in the case of uniform noise-independent generatability, non-uniform noise-independent generatability is still too strong as there exists a class of just two hypotheses that is not non-uniformly noise-independent generatable. We refer the reader to Appendix D for more details.

# 3. Towards Characterizations of Noisy Generation

## 3.1. Uniform Noise-independent Generatability

We start by providing a characterization of the strongest form of noisy generation – uniform noise-independent gen-

eratability.

**Theorem 3.1** (Characterization of Uniform Noise-independent Generatability). *Let $\mathcal{X}$ be countable and $\mathcal{H} \subseteq \{0,1\}^{\mathcal{X}}$ satisfy the* UUS *property. Then, $\mathcal{H}$ is uniformly noise-independent generatable if and only if $\left|\bigcap_{h \in \mathcal{H}} \mathrm{supp}(h)\right| = \infty$.*

Theorem 3.1 is a hardness result – it shows that even finite classes with just two hypotheses may not be uniformly noise-independent generatable. In fact, one way to interpret Theorem 3.1 is that uniform noise-independent generation is only possible for trivial classes where the generator can perfectly generate without observing any examples from the adversary. Indeed, this is what condition in Theorem 3.1 implies, since if this condition is true, the generator can simply compute $\bigcap_{h \in \mathcal{H}} \mathrm{supp}(h)$ and always play from this set. In Appendix C, we prove a similar statement even if we only measure the sample complexity in terms of the number of distinct positive examples. Since the sufficiency direction of Theorem 3.1 follows from this observation, we only prove the necessity direction.

*Proof.* (of necessity in Theorem 3.1) Let $\mathcal{X}$ be countable and $\mathcal{H} \subseteq \{0,1\}^{\mathcal{X}}$ satisfy the UUS property. Suppose that $\left|\bigcap_{h \in \mathcal{H}} \mathrm{supp}(h)\right| =: n < \infty$. We need to show that for every $\mathcal{G}$ and sufficiently large $d \in \mathbb{N}$, there exists a $h \in \mathcal{H}$ and a sequence $x_1, x_2, \ldots$ with $\sum_{t=1}^{\infty} \mathbb{1}\{x_t \notin \mathrm{supp}(h)\} < \infty$, such that for every $t \in \mathbb{N}$ where $|\{x_1, \ldots, x_t\}| = d$, there exists an $s \geq t$ such that $\mathcal{G}(x_{1:s}) \notin \mathrm{supp}(h) \setminus \{x_1, \ldots, x_s\}$. To that end, fix a generator $\mathcal{G}$ and a number $d \geq n$. Let $x_1, \ldots, x_n$ be the sequence of $n$ examples in $\bigcap_{h \in \mathcal{H}} \mathrm{supp}(h)$ sorted in their natural order. Pick any sequence of distinct examples $x_{n+1}, x_{n+2}, \ldots, x_d$ and concatenate them to the end $x_{1:n}$. Let $\hat{x} = \mathcal{G}(x_{1:d})$ and suppose without loss of generality that $\hat{x} \notin \{x_{1:d}\}$. Then, by construction, there exists a $h \in \mathcal{H}$ such that $\hat{x} \notin \mathrm{supp}(h)$. Finally, complete the stream by picking distinct $\{x_{d+1}, x_{d+2}, \ldots\} \subseteq \mathrm{supp}(h)$. First, by construction, note that $\sum_{t=1}^{\infty} \mathbb{1}\{x_t \notin \mathrm{supp}(h)\} \leq d - n < \infty$. Next, note that $t = d$ is the only time point such that $|\{x_{1:d}\}| = d$. Finally, note that when $s = t = d$, we have that $\mathcal{G}(x_{1:s}) = \hat{x} \notin \mathrm{supp}(h) \setminus \{x_{1:s}\}$ by definition. If however $d < n$, then as soon as the common intersection $x_{1:n}$ is enumerated, the generator is forced to produce an output that necessarily lies outside of the support of some hypothesis in $\mathcal{H}$. This completes the proof. $\square$

## 3.2. Uniform Noise-dependent Generatability

Theorem 3.1 shows that obtaining guarantees that are uniform over the noise-level is generally hopeless. In this section, we study the easier setting of uniform noise-dependent generation, where the number of examples needed for perfect generation can depend on the noise-level. At a high

level, the results in this section extend the techniques from Li et al. (2024) to account for noisy example streams. As such, we first define a noisy analog of the Closure dimension (see Appendix B for a definition).

**Definition 3.2** ($n$-Noisy Closure dimension). The *Noisy Closure dimension* of $\mathcal{H}$ at noise-level $n \in \mathbb{N}$, denoted $\mathrm{NC}_n(\mathcal{H})$, is the largest natural number $d$ for which there exist *distinct* $x_1, \ldots, x_d \in \mathcal{X}$ such that $\langle x_1, \ldots, x_d \rangle_{\mathcal{H},n} \neq \perp$ and $|\langle x_1, \ldots, x_d \rangle_{\mathcal{H},n}| < \infty$. If this is true for arbitrarily large $d \in \mathbb{N}$, then we say that $\mathrm{NC}_n(\mathcal{H}) = \infty$. On the other hand, if this is not true for $d = 1$, we say that $\mathrm{NC}_n(\mathcal{H}) = 0$.

Unlike the Closure dimension, the Noisy Closure dimension is *scale-sensitive*, as it is defined with respect to every noise-level $n \in \mathbb{N}$. Scale-sensitive dimensions are not new to learning theory, but have also been defined and used to characterize other properties of hypothesis classes like PAC and online learnability for regression problems (Rakhlin et al., 2015; Bartlett et al., 1994). Our main theorem in this section uses the Noisy Closure dimension to provide a complete characterization of which classes are uniform noise-dependent generatable.

**Theorem 3.3** (Characterization of Uniform Noise-dependent Generatability). *Let $\mathcal{X}$ be countable and $\mathcal{H} \subseteq \{0,1\}^{\mathcal{X}}$ satisfy the UUS property. Then, $\mathcal{H}$ is uniformly noise-dependent generatable if and only if $\mathrm{NC}_n(\mathcal{H}) < \infty$ for all $n \in \mathbb{N}$.*

Our proof of Theorem 3.3 is *constructive*. To prove necessity, we consider the case where there exists a $n \in \mathbb{N}$ such that $\mathrm{NC}_n(\mathcal{H}) = \infty$. Given any generator $\mathcal{G}$ and any number of distinct elements $d \in \mathbb{N}$, we explicitly pick an $h \in \mathcal{H}$ and a valid noisy stream $x_1, x_2, \ldots$ such that $\mathcal{G}$ makes a mistake even after observing $d$ distinct examples. In fact, a simple modification of our proof shows that if $\mathrm{NC}_n(\mathcal{H}) = d$, then any generator $\mathcal{G}$ must observe at least $d$ distinct examples before perfectly generating positive examples when the noise-level is $n$. To prove sufficiency, we explicitly construct a generator $\mathcal{G}$, which only needs to observe $\mathrm{NC}_n(\mathcal{H}) + 1$ distinct examples before being able to perfectly generate when the noise-level is $n \in \mathbb{N}$. *Crucially, our generator $\mathcal{G}$ does not need to know the noise-level picked by the adversary.* Together, our necessity and sufficiency directions show that not only does the finiteness of $\mathrm{NC}_n(\mathcal{H})$ at every $n \in \mathbb{N}$ provide a qualitative characterization of uniform noise-dependent generatability, but it also provides a quantitative one – the "sample complexity" at noise-level $n \in \mathbb{N}$ is $\Theta(\mathrm{NC}_n(\mathcal{H}))$ – analogous to the mistake bound in online learning, which quantifies learnability. We defer the full proof to Appendix E.

Our characterization of uniform noise-dependent generatability in terms of the Noisy Closure dimension allows us to show that all finite classes are uniformly noise-dependent generatable. This contrasts uniform noise-independent gen-

eratable, where even simple classes of size two may not be uniformly noise-independent generatable.

**Corollary 3.4** (All Finite Classes are Uniformly Noise-dependent Generatable). *Let $\mathcal{X}$ be countable and $\mathcal{H} \subseteq \{0,1\}^{\mathcal{X}}$ satisfy the UUS property. If $\mathcal{H}$ is finite, then $\mathcal{H}$ is uniformly noise-dependent generatable.*

*Proof.* Let $\mathcal{X}$ be countable and $\mathcal{H} \subseteq \{0,1\}^{\mathcal{X}}$ be a finite hypothesis class with $q := |\mathcal{H}|$ satisfying the UUS property. To show that $\mathcal{H}$ is uniformly noise-dependent generatable, it suffices to show that $\mathrm{NC}_n(\mathcal{H}) < \infty$ for all $n \in \mathbb{N}$. For any subset $V \subseteq \mathcal{H}$, define $\langle \emptyset \rangle_V := \bigcap_{h \in V} \mathrm{supp}(h)$ and $\mathcal{F} = \{V \subseteq \mathcal{H} : |\langle \emptyset \rangle_V| < \infty\}$ to be the set of examples common to all hypotheses in $V$ and the subsets of $\mathcal{H}$ whose intersection of supports has finite cardinality, respectively. Then, let $d := \max_{V \in \mathcal{F}} |\langle \emptyset \rangle_V|$ be the maximum size of a finite set of examples common to a subset of $\mathcal{H}$. Note that $d$ is finite because $\mathcal{H}$ is finite. Fix any noise-level $n \in \mathbb{N}$ and for the sake of contradiction, suppose $\mathrm{NC}_n(\mathcal{H}) = \infty$. This means that for every $s \in \mathbb{N}$, there exists $t \geq s$ and a sequence of distinct examples $x_1, \ldots, x_t$ such that $|\langle x_1, \ldots, x_t \rangle_{\mathcal{H},n}| < \infty$. Pick $s = nq + d + 1$ and consider the stream $x_1, \ldots, x_t$ such that $|\langle x_1, \ldots, x_t \rangle_{\mathcal{H},n}| < \infty$ for $t \geq s$. Recall that $\mathcal{H}(x_{1:t}; n) = \{h \in \mathcal{H} : |\{x_1, \ldots, x_t\} \cap \mathrm{supp}(h)| \geq t - n\}$. That is, $\mathcal{H}(x_{1:t}; n)$ contains all hypotheses in $\mathcal{H}$ that are inconsistent with $x_{1:t}$ on at most $n$ examples. Since $|\mathcal{H}(x_{1:t}; n)| \leq |\mathcal{H}| = q$, there are at least $t - nq \geq (nq + d + 1) - nq = d + 1$ distinct examples in $x_{1:t}$ contained in the support of all hypotheses in $\mathcal{H}(x_{1:t}; n)$. In other words, $|\langle x_1, \ldots, x_t \rangle_{\mathcal{H},n}| = |\langle \emptyset \rangle_{\mathcal{H}(x_{1:t};n)}| \geq d + 1 > d$. This is a contradiction as $d = \max_{V \in \mathcal{F}} |\langle \emptyset \rangle_V|$ by definition and $\mathcal{H}(x_{1:t}; n) \in \mathcal{F}$. Thus, $\mathrm{NC}_n(\mathcal{H}) < nq + d + 1 < \infty$ for all $n \in \mathbb{N}$, completing the proof. $\square$

We end this section by establishing that uniform noise-dependent generatability is strictly harder than noiseless uniform generatability. This is similar to online learnability for classification problems, where typically the noise-free (realizable) and noisy (agnostic) characterizations of learnability are not equivalent for deterministic learning algorithms (Littlestone, 1987; Ben-David et al., 2009).

**Lemma 3.5** (Uniform Generatability $\neq$ Uniform Noise-dependent Generatability). *Let $\mathcal{X}$ be countable. There exists a countable class $\mathcal{H} \subseteq \{0,1\}^{\mathcal{X}}$ satisfying the UUS property such that $\mathrm{C}(\mathcal{H}) = 0$ but $\mathrm{NC}_1(\mathcal{H}) = \infty$.*

*Proof.* Let $\mathbb{P} = \{p_1, p_2, \ldots\}$ be the set of primes, $S_q = \{-q^n : n \in \mathbb{N}\}$, $\mathbb{S} = \bigcup_{q \in \mathbb{P}} S_q$, and $\mathcal{X} = \mathbb{P} \cup \mathbb{S}$. Let the support of each hypothesis $h_p$ be $\mathrm{supp}(h_p) = \mathbb{P} \backslash \{p\} \cup \mathbb{S} \backslash S_p$ and $\mathcal{H} := \{h_p : p \in \mathbb{P}\}$. Observe that $\mathcal{H}$ satisfies the UUS property and $\mathcal{X}$ is countable.

We now prove that the Closure dimension (see Appendix B) $C(\mathcal{H}) = 0$. Suppose the first element in the sequence is $x_1 = -p^n$ for some $p \in \mathbb{P}$ and $n \in \mathbb{N}$. Then $|\langle x_1 \rangle_{\mathcal{H}}| = |\{-p^m : m \in \mathbb{N}\}| = \infty$. If the first element in the sequence is $x_1 = p \in \mathbb{P}$, then $|\langle x_1 \rangle_{\mathcal{H}}| = |\{-p^n : n \in \mathbb{N}\}| = \infty$. This is because observing any example immediately rules out the one hypothesis that predicts 0 for that example. Note that these are the only two possible cases as it must be the case that $x_1 \in \text{supp}(h)$ for some $h \in \mathcal{H}$.

Now, we show that $\text{NC}_1(\mathcal{H}) = \infty$. Fix some $d \in \mathbb{N}$. Consider the stream $p_1, p_2, \ldots, p_d$ of the first $d$ prime numbers. Then, $|\langle p_1, \ldots, p_d \rangle_{\mathcal{H},1}| = |\emptyset| = 0$, due to the fact that $\mathcal{H}(p_1, \ldots, p_d; 1) = \mathcal{H}$ and the intersection of the supports of all $h \in \mathcal{H}$ is empty. So, $\text{NC}_1(\mathcal{H}) \geq d$. Since $d \in \mathbb{N}$ was picked arbitrarily, this is true for all $d \in \mathbb{N}$ and therefore, $\mathcal{H}$ is not uniformly noise-dependent generatable. $\quad\square$

Lemma 3.5 shows a strong separation between noise-free and uniform noise-dependent generation – there is a class that is uniformly generatable, but not uniformly noise-dependent generatable even when the adversary is allowed to perturb *one* example.

### 3.3. Non-uniform Noise-dependent Generatability

Similar to the characterization of non-uniform generatability in the noise-free setting, we can use uniform noise-dependent generatability to provide sufficiency and necessary conditions for *non-uniform* noise-dependent generatability.

**Lemma 3.6** (Sufficiency for Non-uniform Noise-dependent Generatability). *Let $\mathcal{X}$ be countable and $\mathcal{H} \subseteq \{0,1\}^{\mathcal{X}}$ satisfy the* UUS *property. If there exists a non-decreasing sequence of classes $\mathcal{H}_1 \subseteq \mathcal{H}_2 \subseteq \cdots$ such that $\mathcal{H} = \bigcup_{i=1}^{\infty} \mathcal{H}_i$ and $\text{NC}_i(\mathcal{H}_i) < \infty$ for all $i \in \mathbb{N}$, then $\mathcal{H}$ is non-uniformly noise-dependent generatable.*

To prove Lemma 3.6, we construct a non-uniform noise-dependent generator $\mathcal{G}$, which at each round $t \in \mathbb{N}$, computes an index $i_t \in \mathbb{N}$ based on the number of distinct examples seen in the stream so far. $\mathcal{G}$ then computes the noisy closure of $\mathcal{H}_{i_t}$ at noise-level $i_t$ and then plays from this set. We defer details to Appendix F.

As a corollary of Lemma 3.6 and Corollary 3.4, we can show that all countable classes are non-uniformly noise-dependent generatable, and hence noisily generatable in the limit. In some sense, this result shows that amongst countable classes, noisy generation comes for free!

**Corollary 3.7** (All Countable Classes are Noisily Non-uniformly Generatable). *Let $\mathcal{X}$ be countable and $\mathcal{H} \subseteq \{0,1\}^{\mathcal{X}}$ satisfy the* UUS *property. If $\mathcal{H}$ is countable, then $\mathcal{H}$ is non-uniformly noise-dependent generatable, and therefore also noisily generatable in the limit.*

*Proof.* Since non-uniform noise-dependent generatability implies noisy generatability in the limit, it suffices to show that every countable $\mathcal{H}$ is non-uniformly noise-dependent generatable. Let $\mathcal{X}$ be countable and $\mathcal{H} \subseteq \{0,1\}^{\mathcal{X}}$ be any countable class satisfying the UUS property. Let $h_1, h_2, \ldots$ be some fixed enumeration of $\mathcal{H}$ and define $\mathcal{H}_n = \{h_i : i \leq n\}$ for all $n \in \mathbb{N}$. Then, observe that $\mathcal{H}_1 \subseteq \mathcal{H}_2 \subseteq \ldots$ and that $\mathcal{H} = \bigcup_{n=1}^{\infty} \mathcal{H}_n$. Next, observe that $|\mathcal{H}_n| = n < \infty$, which, using Corollary 3.4, implies that $\text{NC}_n(\mathcal{H}_n) < \infty$. Finally, Lemma 3.6 gives that $\mathcal{H}$ is non-uniformly noise-dependent generatable. $\quad\square$

Corollary 3.7 with Lemma 3.5 also establishes that uniform noise-dependent generatability is strictly harder than non-uniform noise-dependent generatability. We now move to our necessity condition for non-uniform noise-dependent generatability.

**Lemma 3.8** (Necessity for Non-uniform Noise-dependent Generatability). *Let $\mathcal{X}$ be countable and $\mathcal{H} \subseteq \{0,1\}^{\mathcal{X}}$ satisfy the* UUS *property. If $\mathcal{H}$ is non-uniformly noise-dependent generatable, then for every $n \in \mathbb{N}$, there exists a non-decreasing sequence of classes $\mathcal{H}_1 \subseteq \mathcal{H}_2 \subseteq \cdots$ such that $\mathcal{H} = \bigcup_{i=1}^{\infty} \mathcal{H}_i$ and $\text{NC}_n(\mathcal{H}_i) < \infty$ for all $i \in \mathbb{N}$.*

*Proof.* Let $\mathcal{X}$ be countable and $\mathcal{H} \subseteq \{0,1\}^{\mathcal{X}}$ be any non-uniformly noise-dependent generatable class satisfying the UUS property. Let $\mathcal{G}$ be a non-uniform noise-dependent generator for $\mathcal{H}$. Fix $n \in \mathbb{N}$. For every $h \in \mathcal{H}$, let $d_{h,n} \in \mathbb{N}$ be the smallest natural number such that for any sequence $x_1, x_2, \ldots$ with $\sum_{t=1}^{\infty} \mathbb{1}\{x_t \notin \text{supp}(h)\} \leq n$, if there exists a $t \in \mathbb{N}$ such that $|\{x_1, \ldots, x_t\}| = d_{h,n}$, then $\mathcal{G}(x_{1:s}) \in \text{supp}(h) \setminus \{x_1, \ldots, x_s\}$ for all $s \geq t$. Let $\mathcal{H}_i = \{h \in \mathcal{H} : d_{h,n} \leq i\}$ for all $i \in \mathbb{N}$. Note that $\mathcal{H} = \bigcup_{i \in \mathbb{N}} \mathcal{H}_i$ because $d_{h,n} < \infty$ for all $h \in \mathcal{H}$. Moreover, we have that $\mathcal{H}_i \subseteq \mathcal{H}_{i+1}$ for all $i \in \mathbb{N}$. Finally, for every $i \in \mathbb{N}$, observe that $\mathcal{G}$ is a uniform generator for $\mathcal{H}_i$ at noise-level $n \in \mathbb{N}$, implying that $\text{NC}_n(\mathcal{H}_i) < i < \infty$. $\quad\square$

Note the sufficiency and necessary conditions in Lemmas 3.6 and 3.8 respectively are *not* matching. Indeed, the condition in Lemma 3.6 is stronger than that in Lemma 3.8. We leave as an open question a complete characterization of non-uniform noise-dependent generatability.

### 3.4. Noisy Generatability in the Limit

Corollary 3.7 showed that all countable classes are noisily generatable in the limit. Here, we provide alternate sufficiency conditions for noisy generatability in the limit. Our first result shows that (noiseless) non-uniform generatability is sufficient for noisy generatability in the limit. Since all countable classes are (noiseless) non-uniform generatable (Li et al., 2024), this result also shows that all countable classes are noisily generatable in the limit.

---

**Algorithm 1** Generator $\mathcal{G}$

---

**Input:** Hypothesis class $\mathcal{H}$ and non-uniform generator $\mathcal{Q}$
**for** $t = 1, 2, \ldots$ **do**
    Adversary reveals example $x_t$
    Let $d_t = |\{x_1, \ldots, x_t\}|$ and $r_t \leq t$ be the largest time point such that $|\{x_{r_t}, \ldots, x_t\}| = \lfloor \frac{d_t}{2} \rfloor$
    Initialize $\hat{z}_1^t = \mathcal{Q}(x_{r_t : t})$
    **for** $i = 1, \ldots, r_t - 1$ **do**
        **if** $\hat{z}_i^t \notin \{x_1, \ldots, x_t\}$ **then**
            $\mathcal{G}$ outputs $\hat{z}_i^t$ and moves to next round.
        **else**
            Update $\hat{z}_{i+1}^t = \mathcal{Q}(x_{r_t}, \ldots, x_t, \hat{z}_1^t, \ldots, \hat{z}_i^t)$
        **end if**
    **end for**
    $\mathcal{G}$ outputs $\hat{z}_{r_t}^t$ and moves to next round.
**end for**

---

**Theorem 3.9** (Non-uniform Generatability $\implies$ Noisy Generatability in the Limit). *Let $\mathcal{X}$ be countable and $\mathcal{H} \subseteq \{0, 1\}^{\mathcal{X}}$ be any class satisfying the* UUS *property. If $\mathcal{H}$ is (noiseless) non-uniformly generatable, then $\mathcal{H}$ is noisily generatable in the limit.*

*Proof.* Let $\mathcal{X}$ be countable and $\mathcal{H} \subseteq \{0, 1\}^{\mathcal{X}}$ be any class satisfying the UUS property. Consider the generator $\mathcal{G}$, which uses a non-uniform generator $\mathcal{Q}$ as in Algorithm 1.

We will show that $\mathcal{G}$ noisily generates from $\mathcal{H}$ in the limit. Let $h^\star \in \mathcal{H}$ and $x_1, x_2, \ldots$ be the chosen hypothesis and noisy enumeration of $\mathrm{supp}(h^\star)$ picked by the adversary, respectively. Let $d_{\mathcal{Q}}(h^\star) \in \mathbb{N}$ be the number of *noiseless* distinct examples that $\mathcal{Q}$ needs for $h^\star$ to perfectly generate from $\mathrm{supp}(h^\star)$. Let $t^\star \in \mathbb{N}$ be such that for all $t \geq t^\star$, we have that $x_t \in \mathrm{supp}(h^\star)$. Such a $t^\star$ must exist because there are at most a finite number of noisy examples in the stream. Also, because $x_1, x_2, \ldots$ is a noisy enumeration, at some point $s^\star \in \mathbb{N}$, $r_{s^\star} \geq t^\star$ and $\lfloor \frac{d_{s^\star}}{2} \rfloor \geq d_{\mathcal{Q}}(h^\star)$. Fix some $s \geq s^\star$. We will prove that $\mathcal{G}$ generates perfectly on round $s$. First, we claim that for all $i \in [r_s]$, we have that $\hat{z}_i^s \in \mathrm{supp}(h^\star) \setminus \{x_{r_s}, \ldots, x_s, \hat{z}_1^s, \ldots, \hat{z}_{i-1}^s\}$. Our proof will be by induction. For the base case, consider $i = 1$. We need to show that $\hat{z}_1^s \in \mathrm{supp}(h^\star) \setminus \{x_{r_s}, \ldots, x_s\}$. However, this just follows from the fact that $\hat{z}_1^s = \mathcal{Q}(x_{r_s : s})$, $|\{x_{r_s}, \ldots, x_s\}| \geq d_{\mathcal{Q}}(h^\star)$, and $\{x_{r_s}, \ldots, x_s\} \subseteq \mathrm{supp}(h^\star)$. Next, for the induction step, suppose that for all $i \leq m$, we have that $\hat{z}_i^s \in \mathrm{supp}(h^\star) \setminus \{x_{r_s}, \ldots, x_s, \hat{z}_1^s, \ldots, \hat{z}_{i-1}^s\}$. Then, by definition, we have that $\hat{z}_{m+1}^s = \mathcal{Q}(x_{r_s}, \ldots, x_s, \hat{z}_1^s, \ldots, \hat{z}_m^s)$. The proof of the claim is complete after noting that $\{x_{r_s}, \ldots, x_s, \hat{z}_1^s, \ldots, \hat{z}_m^s\} \subseteq \mathrm{supp}(h^\star)$ and $|\{x_{r_s}, \ldots, x_s, \hat{z}_1^s, \ldots, \hat{z}_m^s\}| \geq d_{\mathcal{Q}}(h^\star)$.

Now we will complete the overall proof by showing that the output of $\mathcal{G}$ on round $s$ lies in $\mathrm{supp}(h^\star) \setminus \{x_1, \ldots, x_s\}$.

Suppose that $\mathcal{G}$ outputs $\hat{z}_j^s$ for some $j \in [r_s]$. If $j < r_s$, then it must be the case that Line 7 fired and so by construction $\hat{z}_j^s \in \mathrm{supp}(h^\star) \setminus \{x_1, \ldots, x_s\}$. Thus, suppose that $j = r_s$. If $j = r_s$, then it must be the case that $\{\hat{z}_1^s, \ldots, \hat{z}_{r_s-1}^s\} = \{x_1, \ldots, x_{r_s-1}\}$. Accordingly, we have that $\{x_{r_s}, \ldots, x_s, \hat{z}_1^s, \ldots, \hat{z}_{r_s-1}^s\} = \{x_1, \ldots, x_s\}$, which means that $\hat{z}_{r_s}^s \in \mathrm{supp}(h^\star) \setminus \{x_{r_s}, \ldots, x_s, \hat{z}_1^s, \ldots, \hat{z}_{r_s-1}^s\}$ and therefore $\hat{z}_{r_s}^s \in \mathrm{supp}(h^\star) \setminus \{x_1, \ldots, x_s\}$. Thus, in either case, $\mathcal{G}$ perfectly generates on round $s$. Since $s \geq s^\star$ is arbitrary, our proof is complete. $\qquad\square$

In similar spirit to Theorem 3.10 from Li et al. (2024), we provide another sufficiency condition for noisy generatability in the limit in terms of *uniform noise-independent generatability*.

**Theorem 3.10.** *Let $\mathcal{X}$ be countable and $\mathcal{H} \subseteq \{0, 1\}^{\mathcal{X}}$ satisfy the* UUS *property. If there exists a finite sequence of uniformly noise-independent generatable classes $\mathcal{H}_1, \ldots, \mathcal{H}_k$ such that $\mathcal{H} = \bigcup_{i=1}^k \mathcal{H}_i$, then $\mathcal{H}$ is noisily generatable in the limit.*

The proof of Theorem 3.10 is similar to the proof Theorem 3.10 from Li et al. (2024) and so we defer it to Appendix G.

# 4. Discussion and Open Questions

In this paper, we introduced several notions of noisy generatability and made progress towards characterizing which classes are noisily generatable. We highlight some important directions of future work.

**Characterizations of Non-uniform Noise-dependent Generatability and Noisy Generatability in the Limit.** An important direction of future work is to provide a complete and concise characterization of which classes are non-uniformly noise-dependent generatable and noisily generatable in the limit. Note that Li et al. (2024) also pose the complete characterization of (noiseless) generatability in the limit as an open question, but do provide a complete characterization of (noiseless) non-uniform generatability. We conjecture that the correct characterization of non-uniform noise-dependent generatability will need to go beyond our sufficiency and necessity conditions in Section 3.3.

**Noisy Generatability in the Limit via Membership Oracles.** In addition to showing that all countable classes are generatable in the limit, Kleinberg & Mullainathan (2024) provide a computable algorithm for doing so when given access to a *membership oracle*. A membership oracle, $\mathcal{O}_{\mathcal{H}} : \mathcal{H} \times \mathcal{X} \to \{0, 1\}$, takes as input a hypothesis $h \in \mathcal{H}$ and an example $x \in \mathcal{X}$ and outputs $\mathbb{1}\{h(x) = 1\}$. Although we show that all countable classes are also noisily generatable in the limit, the focus of this paper was information-theoretic in nature. This motivates our next open question. Given access to a membership oracle, does

there exists a *computable* algorithm that can noisily generate in the limit from any countable class?

## Acknowledgements

The authors thank anonymous reviewer gxU2 for catching a bug in Theorem 3.1 and providing a fix. VR acknowledges support from the NSF Graduate Research Fellowship Program (GRFP).

## Impact Statement

This paper presents work whose goal is to advance the field of Machine Learning. There are many potential societal consequences of our work, none of which we feel must be specifically highlighted here.

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

# A. Definitions of Noiseless Generation

**Definition A.1** (Uniform Generatability (Li et al., 2024)). Let $\mathcal{H} \subseteq \{0,1\}^{\mathcal{X}}$ be any hypothesis class satisfying the UUS property. Then, $\mathcal{H}$ is *uniformly generatable*, if there exists a generator $\mathcal{G}$ and $d^{\star} \in \mathbb{N}$, such that for every $h \in \mathcal{H}$ and any sequence $x_1, x_2, \ldots$ with $\{x_1, x_2, \ldots\} \subseteq \mathrm{supp}(h)$, if there exists $t^{\star} \in \mathbb{N}$ such that $|\{x_1, \ldots, x_{t^{\star}}\}| = d^{\star}$, then $\mathcal{G}(x_{1:s}) \in \mathrm{supp}(h) \setminus \{x_1, \ldots, x_s\}$ for all $s \geq t^{\star}$.

**Definition A.2** (Non-uniform Generatability (Li et al., 2024)). Let $\mathcal{H} \subseteq \{0,1\}^{\mathcal{X}}$ be any hypothesis class satisfying the UUS property. Then, $\mathcal{H}$ is *non-uniformly generatable* if there exists a generator $\mathcal{G}$ such that for every $h \in \mathcal{H}$, there exists a $d^{\star} \in \mathbb{N}$ such that for any sequence $x_1, x_2, \ldots$ with $\{x_1, x_2, \ldots\} \subseteq \mathrm{supp}(h)$, if there exists $t^{\star} \in \mathbb{N}$ such that $|\{x_1, \ldots, x_{t^{\star}}\}| = d^{\star}$, then $\mathcal{G}(x_{1:s}) \in \mathrm{supp}(h) \setminus \{x_1, \ldots, x_s\}$ for all $s \geq t^{\star}$.

**Definition A.3** (Generatability in the Limit (Li et al., 2024)). Let $\mathcal{H} \subseteq \{0,1\}^{\mathcal{X}}$ be any hypothesis class satisfying the UUS property. Then, $\mathcal{H}$ is *generatable in the limit* if there exists a generator $\mathcal{G}$ such that for every $h \in \mathcal{H}$, and any *enumeration* $x_1, x_2, \ldots$ of $\mathrm{supp}(h)$, there exists a $t^{\star} \in \mathbb{N}$ such that $\mathcal{G}(x_{1:s}) \in \mathrm{supp}(h) \setminus \{x_1, \ldots, x_s\}$ for all $s \geq t^{\star}$.

# B. Summary of Results for Noiseless Generation

In the noiseless setting, Kleinberg & Mullainathan (2024) initiated the study of generation in the limit by showing that all countable classes are generatable in the limit and that all finite classes are uniformly generatable. Following this result, Li et al. (2024) provided a complete characterization of uniform generation in terms of a new combinatorial parameter termed the Closure dimension.

**Definition B.1** (Closure dimension (Li et al., 2024)). The *Closure dimension* of $\mathcal{H}$, denoted $\mathrm{C}(\mathcal{H})$, is the largest natural number $d \in \mathbb{N}$ for which there exists *distinct* $x_1, \ldots, x_d \in \mathcal{X}$ such that $\langle x_1, \ldots, x_d \rangle_{\mathcal{H},0} \neq \perp$ and $|\langle x_1, \ldots, x_d \rangle_{\mathcal{H},0}| < \infty$. If this is true for arbitrarily large $d \in \mathbb{N}$, then we say that $\mathrm{C}(\mathcal{H}) = \infty$. On the other hand, if this is not true for $d = 1$, we say that $\mathrm{C}(\mathcal{H}) = 0$.

In particular, a class is uniformly generatable if and only its its Closure dimension is finite.

**Theorem B.2** (Theorem 3.3. in Li et al. (2024)). *A class* $\mathcal{H} \subseteq \{0,1\}^{\mathcal{X}}$ *is* uniformly generatable *if and only if* $\mathrm{C}(\mathcal{H}) < \infty$.

In addition, Li et al. (2024) defined an intermediate setting termed non-uniform generatability, and provided a complete characterization of which classes are non-uniformly generatable.

**Theorem B.3** (Theorem 3.5 in Li et al. (2024)). *A class* $\mathcal{H} \subseteq \{0,1\}^{\mathcal{X}}$ *is* non-uniformly generatable *if and only if there exists a non-decreasing sequence of classes* $\mathcal{H}_1 \subseteq \mathcal{H}_2 \subseteq \ldots$ *such that* $\mathcal{H} = \bigcup_{i=1}^{\infty} \mathcal{H}_i$ *and* $\mathrm{C}(\mathcal{H}_i) < \infty$ *for every* $i \in \mathbb{N}$.

As a corollary, Li et al. (2024) establish that all countable classes are actually non-uniformly generatable. Charikar & Pabbaraju (2024) also established this result independently. With regards to generatability in the limit, Kleinberg & Mullainathan (2024) prove that all countable classes are generatable in the limit. Li et al. (2024) provide an alternate sufficiency condition in terms of the Closure dimension.

**Theorem B.4** (Theorem 3.10 in Li et al. (2024)). *A class* $\mathcal{H} \subseteq \{0,1\}^{\mathcal{X}}$ *is* generatable in the limit *if there exists a finite sequence of classes* $\mathcal{H}_1, \mathcal{H}_2, \ldots, \mathcal{H}_n$ *such that* $\mathcal{H} = \bigcup_{i=1}^{n} \mathcal{H}_i$ *and* $\mathrm{C}(\mathcal{H}_i) < \infty$ *for all* $i \in [n]$.

In this paper, we provide analogous results for noisy generation in terms of a different combinatorial parameter termed the Noisy Closure dimension.

# C. An Alternate Version of Uniform Noise-independent Generatability

In this section, we consider an alternate (weaker) version of uniform noise-independent generatability by measuring the sample complexity only in terms of the valid positive examples observed.

**Definition C.1** (Alternate Uniform Noise-Independent Generatability). Let $\mathcal{H} \subseteq \{0,1\}^{\mathcal{X}}$ be any hypothesis class satisfying the UUS property. Then, $\mathcal{H}$ is *uniformly noise-independent generatable*, if there exists a generator $\mathcal{G}$ and $d^{\star} \in \mathbb{N}$, such that for every $h \in \mathcal{H}$ and any sequence $x_1, x_2, \ldots$ with $\sum_{t=1}^{\infty} \mathbb{1}\{x_t \notin \mathrm{supp}(h)\} < \infty$, if there exists $t^{\star} \in \mathbb{N}$ such that $|\{x_1, \ldots, x_{t^{\star}}\} \cap \mathrm{supp}(h)| = d^{\star}$, then $\mathcal{G}(x_{1:s}) \in \mathrm{supp}(h) \setminus \{x_1, \ldots, x_s\}$ for all $s \geq t^{\star}$.

Note that the only difference between Definition C.1 and D.1 is how $d^{\star}$ is measured. In Definition C.1, $d^{\star}$ captures only the number of distinct positive examples in the stream, while $d^{\star}$ in Definition D.1 measures the total number of distinct examples

in the stream. We start off by proving a simple necessity condition for our alternate notion of uniform noise-independent generatability.

**Lemma C.2** (Necessary condition for Alternate Uniform Noise-independent Generatability)**.** *Let $\mathcal{X}$ be countable and $\mathcal{H} \subseteq \{0,1\}^{\mathcal{X}}$ satisfy the* UUS *property. If there exists a subclass $\mathcal{F} \subseteq \mathcal{H}$ and a hypothesis $f \in \mathcal{F}$ such that $\left|\bigcap_{h \in \mathcal{F}} \operatorname{supp}(h)\right| < \infty$ and $\left|\bigcap_{h \in \mathcal{F} \setminus \{f\}} \operatorname{supp}(h)\right| = \infty$, then $\mathcal{H}$ is* not *uniformly noise-independent generatable according to Definition C.1.*

Like Theorem 3.1, Lemma C.2 is a hardness result – it shows that finite classes with just two hypotheses may not be uniformly noise-independent generatable even when the sample complexity is measured in terms of positive example.

*Proof.* (of Lemma C.2) Let $\mathcal{X}$ be countable and $\mathcal{H} \subseteq \{0,1\}^{\mathcal{X}}$ satisfy the UUS property. Let $\mathcal{F} \subseteq \mathcal{H}$ be any subset of $\mathcal{H}$ such that $\left|\bigcap_{h \in \mathcal{F}} \operatorname{supp}(h)\right| < \infty$ and for which there exists an $f \in \mathcal{F}$ such that $\left|\bigcap_{h \in \mathcal{F} \setminus \{f\}} \operatorname{supp}(h)\right| = \infty$. We need to show that $\mathcal{H}$ is not uniformly noise-independent generatable. We will show that $\mathcal{F}$ is not uniformly noise-independent generatable which will imply that $\mathcal{H}$ is not uniformly noise-independent generatable. To that end, we need to show that for every $\mathcal{G}$ and any $d \in \mathbb{N}$, there exists a $h \in \mathcal{F}$ and a sequence $x_1, x_2, \ldots$ with $\sum_{t=1}^{\infty} \mathbb{1}\{x_t \notin \operatorname{supp}(h)\} < \infty$, such that for every $t \in \mathbb{N}$ where $|\{x_1, \ldots, x_t\} \cap \operatorname{supp}(h)| = d$, there exists an $s \geq t$ such that $\mathcal{G}(x_{1:s}) \notin \operatorname{supp}(h) \setminus \{x_1, \ldots, x_s\}$. To that end, fix a generator $\mathcal{G}$ and a number $d \in \mathbb{N}$. Let $x_1, \ldots, x_d$ be a sequence of $d$ unique examples in $\bigcap_{h \in \mathcal{F} \setminus \{f\}} \operatorname{supp}(h)$. Let $z_1, \ldots, z_d$ be any $d$ unique examples in $\operatorname{supp}(f) \setminus \bigcap_{h \in \mathcal{F}} \operatorname{supp}(h)$ and consider the sequence of $2d$ unique examples obtained by concatenating $z_1, \ldots, z_d$ to the end of $x_1, \ldots, x_d$. Denote this new sequence by $x_{1:2d}$. Let $q = |\bigcap_{h \in \mathcal{F}} \operatorname{supp}(h)|$ and let $v_1, \ldots, v_q$ be its elements sorted in its natural order. Concatenate $v_1, \ldots, v_q$ to the end of $x_{1:2d}$ and denote this new sequence of unique examples as $x_{1:2d+q}$. Observe that for every $h \in \mathcal{F}$, we have that $|\operatorname{supp}(h) \cap \{x_1, \ldots x_{2d+q}\}| \geq d$. Let $\hat{x}_{2d+q} = \mathcal{G}(x_1, \ldots, x_{2d+q})$ and suppose without loss of generality that $\hat{x}_{2d+q} \notin \{x_1, \ldots, x_{2d+q}\}$. Let $h^{\star} \in \mathcal{F}$ be a be hypothesis such that $\hat{x}_{2d+q} \notin \operatorname{supp}(h^{\star})$. Such a hypothesis must exist because $\hat{x}_{2d+q} \notin \{x_1, \ldots, x_{2d+q}\}$ and $\bigcap_{h \in \mathcal{F}} \operatorname{supp}(h) \subseteq \{x_1, \ldots, x_{2d+q}\}$. Let $x_{2d+q+1}, x_{2d+q+2}, \ldots$ be any completion of the stream such that $\{x_{2d+q+t}\}_{t=1}^{\infty} \subseteq \operatorname{supp}(h^{\star})$ and $\{x_{2d+q+t}\}_{t=1}^{\infty} \cap \{x_t\}_{t=1}^{2d+q} = \emptyset$.

We are now ready to complete the proof. Let $h^{\star}$ and $x_1, x_2, \ldots$ be the hypothesis and sequence chosen above. Then, by definition, observe that $\sum_{t=1}^{\infty} \mathbb{1}\{x_t \notin \operatorname{supp}(h^{\star})\} = \sum_{t=1}^{2d+q} \mathbb{1}\{x_t \notin \operatorname{supp}(h^{\star})\} \leq d < \infty$. Moreover, we also know that $|\operatorname{supp}(h^{\star}) \cap \{x_1, \ldots, x_{2d+q}\}| \geq d$ and the first $2d$ examples are distinct, implying that there exists exactly one time point $t \leq 2d + q$ such that $|\operatorname{supp}(h^{\star}) \cap \{x_1, \ldots x_{2d+q}\}| = d$. Finally, noting that $2d + q \geq t$ and that $\mathcal{G}(x_1, \ldots, x_{2d+q}) \notin \operatorname{supp}(h^{\star}) \setminus \{x_1, \ldots, x_{2d+1}\}$ completes the proof that $\mathcal{F}$ is not uniformly noise-independent generatable since $\mathcal{G}$ and $d$ were chosen arbitrarily. $\square$

Finally, Theorem C.3 provides a full characterization of uniform noise-independent generatability (Definition D.1 in terms of the noisy closure dimension.

**Theorem C.3** (Characterization of Uniform Noise-independent Generatability)**.** *Let $\mathcal{X}$ be countable and $\mathcal{H} \subseteq \{0,1\}^{\mathcal{X}}$ satisfy the* UUS *property. Then, $\mathcal{H}$ is uniformly noise-independent generatable if and only if $\sup_n (\operatorname{NC}_{n \in \mathbb{N}}(\mathcal{H}) - n) < \infty$.*

As the proof follows similar techniques as those in the main text, we only provide the proof sketch here.

*Proof.* (sketch of Theorem C.3) For the necessity direction, suppose that $\sup_n (\operatorname{NC}_n(\mathcal{H}) - n) = \infty$. Then for every $d \in \mathbb{N}$, we can find a $t \geq d$ and a sequence $x_1, \ldots, x_t$, such that $|\langle x_1, ..., x_t \rangle_{\mathcal{H}, t-d}| < \infty$. Hence, by padding $x_1, \ldots, x_t$ with any remaining elements in $\langle x_1, ..., x_t \rangle_{\mathcal{H}, t-d}$, we can force the Generator to make a mistake while ensuring that the hypothesis chosen is consistent with at least $d$ examples in the stream. For the sufficiency direction, if $\sup_n (\operatorname{NC}_n(\mathcal{H}) - n) < \infty$, then there exists a $d \in \mathbb{N}$ such that for every $t \geq d$ and distinct $x_1, ..., x_t$, we have that either $\langle x_1, ..., x_t \rangle_{\mathcal{H}, t-d} = \bot$ or $|\langle x_1, ..., x_t \rangle_{\mathcal{H}, t-d}| = \infty$. Thus, the algorithm which for $t \geq d$ plays from $\langle x_1, ..., x_t \rangle_{\mathcal{H}, t-d} \setminus \{x_1, \ldots, x_t\}$ if $\langle x_1, ..., x_t \rangle_{\mathcal{H}, t-d} \neq \bot$ is guaranteed to succeed. $\square$

# D. The Hardness of Non-uniform Noise-independent Generatability

As alluded to in Remark 2.8, there is a fifth setting of noisy generation, which we did not define in the main text. In this setting, the "sample complexity" of the generator can be non-uniform over the class $\mathcal{H}$ but must still be uniform over the noise-level and stream picked by the adversary. We define this formally below.

**Definition D.1** (Non-uniform Noise-independent Generatability). Let $\mathcal{H} \subseteq \{0,1\}^{\mathcal{X}}$ be any hypothesis class satisfying the UUS property. Then, $\mathcal{H}$ is *non-uniformly noise-independent generatable*, if there exists a generator $\mathcal{G}$ such that for every $h \in \mathcal{H}$, there exists $d^{\star} \in \mathbb{N}$ such that for any sequence $x_1, x_2, \ldots$ with

$$\sum_{t=1}^{\infty} \mathbb{1}\{x_t \notin \mathrm{supp}(h)\} < \infty,$$

if there exists $t^{\star} \in \mathbb{N}$ such that $|\{x_1, \ldots, x_{t^{\star}}\} \cap \mathrm{supp}(h)| = d^{\star}$, then $\mathcal{G}(x_{1:s}) \in \mathrm{supp}(h) \setminus \{x_1, \ldots, x_s\}$ for all $s \geq t^{\star}$.

Unfortunately, non-uniform noise-independent generatability is still a restrictive property as not all finite classes are non-uniformly noise-independent generatable. This is in contrast to uniform noise-dependent generatability as shown by Corollary 3.4.

**Lemma D.2** (Not All Finite Classes are Non-uniformly Noise-independent Generatable). *Let $\mathcal{X}$ be countable. There exists a finite class $\mathcal{H} \subseteq \{0,1\}^{\mathcal{X}}$ which satisfies the UUS property that is not non-uniformly noise-independent generatable.*

*Proof.* Let $\mathcal{X} = \mathbb{N}$ and consider the class $\mathcal{H} = \{h_e, h_o\}$ such that $h_e(x) = \mathbb{1}\{x \text{ is even}\}$ and $h_o(x) = \mathbb{1}\{x \text{ is odd}\}$. We will show that $\mathcal{H}$ is not non-uniformly noise-independent generatable. We need to show that for every generator $\mathcal{G}$, there exists a hypothesis $h \in \mathcal{H}$ such that for every $d \in \mathbb{N}$, there exists a sequence $x_1, x_2, \ldots$ with

$$\sum_{t=1}^{\infty} \mathbb{1}\{x_t \notin \mathrm{supp}(h)\} < \infty,$$

so that for every $t \in \mathbb{N}$ such that $|\{x_1, \ldots, x_t\} \cap \mathrm{supp}(h)| = d$, there exists a $s \geq t$ where $\mathcal{G}(x_{1:s}) \notin \mathrm{supp}(h) \setminus \{x_1, \ldots, x_s\}$. To that end, fix a generator $\mathcal{G}$. Let $S = \{\mathcal{G}(1, 2, \ldots, i)\}_{i \in \mathbb{N}}$. There are three cases to consider: (1) the number of even numbers in $S$ is finite (2) the number of odd numbers in $S$ is finite and (3) there are infinite number of both even and odd numbers in $S$. Cases (1) and (2) are symmetric. So, without loss of generality, it suffices only to consider Cases (1) and (3).

Starting with Case (1), let $p \in \mathbb{N}$ be the smallest time point such that for all $t \geq p$, we have that $\mathcal{G}(1, \ldots, t)$ is odd. We will pick $h_e$ to use for the lower bound. Fix some $d \in \mathbb{N}$. Consider the sequence $x_1, x_2, \ldots$ such that $x_i = i$ for all $i \leq \max\{p, 2d\}$ and $x_i = 2i$ for all $i > \max\{p, 2d\}$. Note that

$$\sum_{t=1}^{\infty} \mathbb{1}\{x_t \notin \mathrm{supp}(h_e)\} \leq \frac{\max\{p, 2d\}}{2} < \infty.$$

Suppose $2d < p$. Then, observe that only on time point $t = 2d$ do we have that $|\{x_1, \ldots, x_t\} \cap \mathrm{supp}(h_e)| = d$. However, by definition, on time point $s = p \geq 2d = t$, we have that $\mathcal{G}(x_1, \ldots, x_s)$ is odd and therefore not in $\mathrm{supp}(h_e)$. Suppose that $2d \geq p$. Then, like before, only on time point $t = 2d$ do we have that $|\{x_1, \ldots, x_t\} \cap \mathrm{supp}(h_e)| = d$. However, by definition, also on time point $s = 2d \geq p$, we have that $\mathcal{G}(x_1, \ldots, x_s)$ is odd and therefore not in $\mathrm{supp}(h_e)$. Since $d \in \mathbb{N}$ is arbitrary, this is true for all $d \in \mathbb{N}$, completing the proof for Case (1). As mentioned before, the proof for Case (2) follows by symmetry.

We now consider Case (3). For this case, we will pick $h_e$ for the lower bound. Fix some $d \in \mathbb{N}$. Let $p \geq 2d$ be the smallest time point after and including time point $2d$ such that $\mathcal{G}(1, 2, \ldots, p)$ is odd. Such a $p \in \mathbb{N}$ must exist since $S$ contains an infinite number of odd numbers. Now, consider the sequence $x_1, x_2, \ldots$ such that $x_i = i$ for $i \leq p$ and $x_i = 2i$ for all $i > p$. Note that

$$\sum_{t=1}^{\infty} \mathbb{1}\{x_t \notin \mathrm{supp}(h_e)\} \leq \frac{p}{2} < \infty.$$

Observe that only at time point $t = 2d$ do we have that $|\{x_1, \ldots, x_t\} \cap \mathrm{supp}(h_e)| = d$. However, by definition, also on time point $s = p \geq 2d = t$, we have that $\mathcal{G}(x_1, \ldots, x_p)$ is odd and therefore does not lie in $\mathrm{supp}(h_e)$. Since $d \in \mathbb{N}$ is arbitrary, this is true for all $d \in \mathbb{N}$, completing the proof for Case (3) and the overall proof. □

# E. Proof of Theorem 3.3

*Proof.* (of necessity in Theorem 3.3) Let $\mathcal{X}$ be countable and $\mathcal{H} \subseteq \{0,1\}^{\mathcal{X}}$ satisfy the UUS property. Suppose there exists a $n \in \mathbb{N}$ such that $\mathrm{NC}_n(\mathcal{H}) = \infty$. We need to show that this means that $\mathcal{H}$ is not uniformly noise-dependent generatable. In particular, we need to show that for every generator $\mathcal{G}$, there exists a noise level $n^{\star} \in \mathbb{N}$ such that for every $d \in \mathbb{N}$, there exists a hypothesis $h \in \mathcal{H}$ and a sequence $x_1, x_2, \ldots$ with $\sum_{t=1}^{\infty} \mathbb{1}\{x_t \notin \mathrm{supp}(h)\} \leq n^{\star}$, so that for every $t \in \mathbb{N}$ with $|\{x_1, \ldots, x_t\}| = d$, there exists $s \geq t$ where $\mathcal{G}(x_{1:s}) \notin \mathrm{supp}(h) \setminus \{x_1, \ldots, x_s\}$. To that end, let $\mathcal{G}$ be any generator and consider the noise level $n^{\star} = n$ so that $\mathrm{NC}_{n^{\star}}(\mathcal{H}) = \infty$. Fix any $d \in \mathbb{N}$. We will now pick such a hypothesis $h^{\star} \in \mathcal{H}$ and a valid stream $x_1, x_2, \ldots$.

Since $\mathrm{NC}_{n^{\star}}(\mathcal{H}) = \infty$, we know there exists a $d^{\star} \geq d$ and a sequence of distinct examples $x_1, \ldots, x_{d^{\star}}$ such that $\langle x_1, \ldots, x_{d^{\star}}\rangle_{\mathcal{H}, n^{\star}} \neq \perp$ and $|\langle x_1, \ldots, x_{d^{\star}}\rangle_{\mathcal{H}, n^{\star}}| < \infty$. Let $q = |\langle x_1, \ldots, x_{d^{\star}}\rangle_{\mathcal{H}, n^{\star}} \setminus \{x_1, \ldots, x_{d^{\star}}\}|$ and $z_1, \ldots, z_q$ be the elements of $\langle x_1, \ldots, x_{d^{\star}}\rangle_{\mathcal{H}, n^{\star}} \setminus \{x_1, \ldots, x_{d^{\star}}\}$ sorted in their natural order. Let $x_{1:d^{\star}+q}$ denote the sequence obtained by concatenating $z_1, \ldots, z_q$ to the end of $x_1, \ldots, x_{d^{\star}}$. Let $\hat{x}_{d^{\star}+q} = \mathcal{G}(x_{1:d^{\star}+q})$ and suppose without loss of generality that $\hat{x}_{d^{\star}+q} \notin \{x_1, \ldots, x_{d^{\star}+q}\}$. Let $h^{\star} \in \mathcal{H}(x_{1:d^{\star}}, n^{\star})$ be any hypothesis such that $\hat{x}_{d^{\star}+q} \notin \mathrm{supp}(h^{\star})$. Such a hypothesis must exist because $\hat{x}_{d^{\star}+q} \notin \{x_1, \ldots, x_{d^{\star}+q}\}$ and $\langle x_1, \ldots, x_{d^{\star}}\rangle_{\mathcal{H}, n^{\star}} \subseteq \{x_1, \ldots, x_{d^{\star}+q}\}$. Finally, let $x_{d^{\star}+q+1}, x_{d^{\star}+q+2}, \ldots$ be any completion of the stream such that $\{x_{d^{\star}+q+t}\}_{t=1}^{\infty} \subseteq \mathrm{supp}(h^{\star})$ and $\{x_{d^{\star}+q+t}\}_{t=1}^{\infty} \cap \{x_t\}_{t=1}^{d^{\star}+q} = \emptyset$.

We now complete the proof. Let $h^{\star}$ and $x_1, x_2, \ldots$ be the hypothesis and stream chosen above. First, observe that

$$\sum_{t=1}^{\infty} \mathbb{1}\{x_t \notin \mathrm{supp}(h^{\star})\} = \sum_{t=1}^{d^{\star}} \mathbb{1}\{x_t \notin \mathrm{supp}(h^{\star})\} \leq n^{\star}.$$

Second, there exists only one $t \in \mathbb{N}$, namely $t = d$, such that $|\{x_1, \ldots, x_t\}| = d$, as the first $d^{\star} + 1 > d$ examples of the stream are distinct by construction. However, observe that by construction, at time point $s = d^{\star} + q \geq d$, we have that $\mathcal{G}(x_1, \ldots, x_s) = \hat{x}_{d^{\star}+q} \notin \mathrm{supp}(h^{\star}) \setminus \{x_1, \ldots, x_s\}$ by choice of $h^{\star}$. Since $\mathcal{G}$ and $d \in \mathbb{N}$ were picked arbitrarily, this is true for all $\mathcal{G}$ and $d \in \mathbb{N}$, which completes the proof. $\square$

*Proof.* (of sufficiency in Theorem 3.3) The key intuition is that the generator, without knowing the adversary's noise level, continually computes the largest noise level for which it has observed enough unique examples to generate perfectly. Eventually, its calculated noise level will surpass and thus account for the adversary's noise level, ensuring that the generator is perfect from then on.

Let $\mathcal{X}$ be countable and $\mathcal{H} \subseteq \{0,1\}^{\mathcal{X}}$ satisfy the UUS property. We need to show that if $\mathrm{NC}_n(\mathcal{H}) < \infty$ for all $n \in \mathbb{N}$, then $\mathcal{H}$ is uniformly noise-dependent generatable. For a sequence $x_1, x_2, \ldots$, let $d_t := |\{x_1, \ldots, x_t\}|$ denote the number of distinct examples amongst the first $t$ examples. Consider the following generator $\mathcal{G}$. At time $t \in \mathbb{N}$, $\mathcal{G}$ computes $n_t := \max\{n \in [t] : \mathrm{NC}_n(\mathcal{H}) < d_t\} \cup \{0\}$. If $n_t = 0$, $\mathcal{G}$ plays any $x \in \mathcal{X}$. Otherwise, it plays from $\langle x_1, \ldots, x_t\rangle_{\mathcal{H}, n_t} \setminus \{x_1, x_2, \ldots, x_t\}$. We now show that $\mathcal{G}$ is a uniform noise-dependent generator. To that end, let $n^{\star} \in \mathbb{N}$ be the noise level chosen by the adversary. We claim that $\mathcal{G}$ can perfectly generate after observing $\max\{\mathrm{NC}_{n^{\star}}(\mathcal{H}) + 1, n^{\star}\}$ distinct examples. To see why, let $h \in \mathcal{H}$ be any hypothesis, and $x_1, x_2, \ldots$ be any stream such that $\sum_{t=1}^{\infty} \mathbb{1}\{x_t \notin \mathrm{supp}(h)\} \leq n^{\star}$. Without loss of generality, suppose there exists $t^{\star} \in \mathbb{N}$ such that $d_{t^{\star}} = \max\{\mathrm{NC}_{n^{\star}}(\mathcal{H}) + 1, n^{\star}\}$. Observe that $t^{\star} \geq n^{\star}$. Fix any $s \geq t^{\star}$. By definition of $n_t$, it must be the case that $n_s \geq n^{\star}$. Thus, $h \in \mathcal{H}(x_1, \ldots, x_s, n_s)$ and $\langle x_1, \ldots, x_s\rangle_{\mathcal{H}, n_s} \subseteq \mathrm{supp}(h)$. Moreover, because $d_s > \mathrm{NC}_{n_s}(\mathcal{H})$, again by definition, it must be the case that $|\langle x_1, \ldots, x_s\rangle_{\mathcal{H}, n_s}| = \infty$. Accordingly, $\langle x_1, \ldots, x_s\rangle_{\mathcal{H}, n_s} \setminus \{x_1, \ldots, x_s\} \neq \emptyset$ and $\mathcal{G}$ is guaranteed to output an example in $\mathrm{supp}(h)$ on round $s$. Since $s \geq t^{\star}$ was chosen arbitrarily, this is true for all such $s$, completing the proof. $\square$

# F. Proof of Lemma 3.6

*Proof.* Let $\mathcal{X}$ be countable and $\mathcal{H} \subseteq \{0,1\}^{\mathcal{X}}$ be any class satisfying the UUS property. Suppose there exists a non-decreasing sequence of classes $\mathcal{H}_1 \subseteq \mathcal{H}_2 \subseteq \cdots$ such that $\mathcal{H} = \bigcup_{i=1}^{\infty} \mathcal{H}_i$ and $\mathrm{NC}_i(\mathcal{H}_i) < \infty$ for all $i \in \mathbb{N}$. For a sequence $x_1, x_2, \ldots$, let $d_t := |\{x_1, \ldots, x_t\}|$ denote the number of distinct examples amongst the first $t$ examples. Consider the following generator $\mathcal{G}$. At time $t \in \mathbb{N}$, $\mathcal{G}$ computes $j_t := \max\{i \in [t] : \mathrm{NC}_i(\mathcal{H}_i) < d_t\} \cup \{0\}$. If $j_t = 0$, $\mathcal{G}$ plays any $x \in \mathcal{X}$. Otherwise, it plays from $\langle x_1, \ldots, x_t\rangle_{\mathcal{H}_{j_t}, j_t} \setminus \{x_1, x_2, \ldots, x_t\}$. We claim that $\mathcal{G}$ is a non-uniform noise-dependent generator for $\mathcal{H}$. To see why, let $n^{\star} \in \mathbb{N}$ and $h \in \mathcal{H}$ be the noise level and hypothesis picked by the adversary. Let $i^{\star} \in \mathbb{N}$ be the smallest index such that $h \in \mathcal{H}_{i^{\star}}$ and define $j^{\star} = \max\{i^{\star}, n^{\star}\}$. We claim that $\mathcal{G}$ perfectly generates after observing $\max\{\mathrm{NC}_{j^{\star}}(\mathcal{H}_{j^{\star}}) + 1, j^{\star}\}$ distinct examples. To that end, let $x_1, x_2, \ldots$ be any

sequence such that $\sum_{t=1}^{\infty} \mathbb{1}\{x_t \notin \mathrm{supp}(h)\} \le n^\star$. Without loss of generality, suppose there exists $t^\star \in \mathbb{N}$ such that $d_{t^\star} = \max\{\mathrm{NC}_{j^\star}(\mathcal{H}_{j^\star}) + 1, j^\star\}$. Observe that $t^\star \ge j^\star$. Fix any $s \ge t^\star$. By definition of $j_t$, it must be the case that $j_s \ge j^\star \ge n^\star$. Moreover, because $j_s \ge i^\star$, we also have that $h \in \mathcal{H}_{j_s}$. Together, this means that $h \in \mathcal{H}_{j_s}(x_1, \ldots, x_s, j_s)$ and $\langle x_1, \ldots, x_s \rangle_{\mathcal{H}_{j_s}, j_s} \subseteq \mathrm{supp}(h)$. Also, because $d_s > \mathrm{NC}_{j_s}(\mathcal{H}_{j_s})$ it must be the case that $|\langle x_1, \ldots, x_s \rangle_{\mathcal{H}_{j_s}, j_s}| = \infty$. Accordingly, $\langle x_1, \ldots, x_s \rangle_{\mathcal{H}_{j_s}, j_s} \setminus \{x_1, \ldots, x_s\} \neq \emptyset$ and $\mathcal{G}$ is guaranteed to output an example in $\mathrm{supp}(h)$ on round $s$. Since $s \ge t^\star$ was chosen arbitrarily, this is true for all such $s$, completing the proof. $\qquad \square$

## G. Proof of Theorem 3.10

*Proof.* Let $\mathcal{X}$ be countable and $\mathcal{H} \subseteq \{0, 1\}^{\mathcal{X}}$ satisfy the UUS property. Suppose there exists a finite sequence of uniformly noise-independent generatable classes $\mathcal{H}_1, \mathcal{H}_2, \ldots, \mathcal{H}_k$ such that $\mathcal{H} = \bigcup_{i=1}^{k} \mathcal{H}_i$. By Theorem 3.1, we know that for all $i \in [k]$, we have that $\left| \bigcap_{h \in \mathcal{H}_i} \mathrm{supp}(h) \right| = \infty$. For every $i \in [k]$, let $z_1^i, z_2^i, \ldots$ denote the elements of $\bigcap_{h \in \mathcal{H}_i} \mathrm{supp}(h)$ sorted in their natural ordering. Consider the following generator $\mathcal{G}$. Given any noisy enumeration $x_1, x_2, \ldots$ and any $t \in \mathbb{N}$, $\mathcal{G}$ first computes

$$p_t^i := \max\{p \in \mathbb{N} : \{z_1^i, \ldots, z_p^i\} \subseteq \{x_1, \ldots, x_t\}\}$$

for every $i \in [k]$. Then, $\mathcal{G}$ computes $i_t = \arg\max_{i \in [k]} p_t^i$. Finally, $\mathcal{G}$ plays arbitrarily from $\{z_1^{i_t}, z_2^{i_t}, \ldots\} \setminus \{x_1, \ldots x_t\}$. We claim that $\mathcal{G}$ generates from $\mathcal{H}$ in the limit.

To see why, let $h \in \mathcal{H}$ and $x_1, x_2, \ldots$ be the hypothesis and noisy enumeration selected by the adversary. Let $i^\star \in [k]$ be such that $h \in \mathcal{H}_{i^\star}$ and $s^\star \in \mathbb{N}$ be such that for all $t \ge s^\star$, we have that $x_t \in \mathrm{supp}(h)$. Such an $s^\star$ exists because $\sum_{t=1}^{\infty} \mathbb{1}\{x_t \notin \mathrm{supp}(h)\} < \infty$. Let $S^\star \subseteq [k]$ be such that $i \in S^\star$ if and only if $\{z_1^i, z_2^i, \ldots\} \subseteq \{x_1, x_2, \ldots\}$. Note that $i^\star \in S^\star$ by definition of $z_1^{i^\star}, z_2^{i^\star}, \ldots$ and the fact that $x_1, x_2, \ldots$ is a noisy enumeration. By definition of $S^\star$ and $p_t^i$ it must be the case that for all $j \notin S^\star$, we have that $\sup_{t \in \mathbb{N}} p_t^j < \infty$. Since $k < \infty$, we then get that $\max_{j \notin S^\star} \sup_{t \in \mathbb{N}} p_t^j < \infty$. On the other hand, for all $i \in S^\star$, we have that $p_t^i \to \infty$. More precisely, for every $i \in S^\star$, there exists a finite $t_i \in \mathbb{N}$ such that $p_{t_i}^i \ge \max_{j \notin S^\star} \sup_{t \in \mathbb{N}} p_t^j$. Since $|S^\star| < \infty$, we also have that $t^\star := \max_{i \in S^\star} t_i < \infty$. Our proof is complete after noting that for all $t \ge t^\star$, we have that $i_t \in S^\star$ and $\{z_1^{i_t}, z_2^{i_t}, \ldots\} \setminus \{x_1, \ldots x_t\} \subseteq \mathrm{supp}(h)$. $\qquad \square$

