# OpenReview forum: "Generation from Noisy Examples"
_ICML.cc/2025/Conference — ICML 2025 poster_

### Official Review · Reviewer_gxU2 · 2025-03-13

**Overall Recommendation:** 4

**Summary:**

This paper studies the model of language generation in the limit, first introduced by Kleinberg and Mulainathan at NeurIPS 2024 and later extended by Lee et al. It builds on seminal work by Gould and Angwin from the 1960s, which has had a profound influence on learning theory. The study this language generation model is nascent and has several open problems. This work tackles a fundamental open problem: learning from adversarially injected noise. Specifically, the authors consider a scenario in which an adversary inserts a finite number of incorrect examples into the learner’s data stream.

The paper defines several models of language generation that mirror the three frameworks proposed by Lee et al.: uniform generation, non-uniform generation, and generation. Additionally, the authors introduce two further models that require the learner’s number of mistakes to be independent of the noise level. For the primary three models, they characterize conditions under which uniform and non-uniform generation can be achieved in a noisy setting and provide several sufficient conditions for generation with noisy examples. Notably, these conditions demonstrate that noisy generation is feasible for all countable collections – a result that strengthens the stark contrast between language identification and language generation, showing that the latter remains possible even in the presence of adversarial corruptions.

The paper’s results are primarily existential or non-computational, relying on the existence of certain oracles. The authors leave open the challenge of extending these findings to computable or algorithmic results and also leave the complete characterization of generation with noisy examples to future work.

## Update after the rebuttal
I thank the reviewer's for explaining their fix to Theorem 3.1. I maintain my original rating.

**Claims And Evidence:**

The paper’s claims are generally well-supported by formal proofs, and overall, I find the arguments convincing and plausible. However, I have some reservations regarding the proof of Theorem 3.1. The proof begins by considering a set F such that the common intersection of all hypotheses in F is finite, and that there exists at least one hypothesis in F whose removal renders the common support infinite. I am not entirely convinced that such a collection F necessarily exists in all cases. For instance, consider the hypothesis class $H = \\{\mathbb{Z}_{\geq i} \cup \\{\infty\\}\mid i\in \mathbb{N}\\}$.

A potential fix for the proof could be as follows: First, exhaust the common intersection of H, which is finite by definition. Then, regardless of the uniform bound d, consider the time d+C, where C is the size of the common intersection. Two cases arise: if the learner outputs an element from the common intersection, it must be a mistake since that element appears in the training set; if not, there exists some hypothesis h that does not include the learner’s output. In this scenario, one can select this h as the target hypothesis and treat all outputs prior to it (except those in the common intersection) as noise.

**Essential References Not Discussed:**

Not applicable

**Experimental Designs Or Analyses:**

Not applicable

**Methods And Evaluation Criteria:**

Not applicable

**Other Comments Or Suggestions:**

Suggestions to improve writing
1. Li et al. leave the complete characterization of generation in the limit open, and while this is not immediately relevant to the work, it would be useful to highlight this in the related works for the benefit of the reader. The authors do mention this in section 4 of discussion and open questions, but I believe it also belongs in the related work.
2. In addition, the naming of Assumption 2.1 seems unnecessarily complex. instead of writing it this way, one could for instance, say “Let $H$ be any hypothesis class consisting of infinite hypothesis”
3. The names of Definitions 2.4 and 2.5 are also somewhat confusing and difficult to distinguish. I strongly suggest renaming them to more descriptive terms like “noise level independent uniform generatability” and “noise level dependent uniform generatability.”
4. Moreover, the paper alternates between interpreting hypotheses as functions and as sets. It would be helpful for the reader to be consistent about how hypotheses are interpreted as functions or as sets.
5. Finally, given the number of interesting results, it would be very useful to include a diagram that summarizes the different notions of generation (both with and without noise) along with their corresponding dimensions. Arrows illustrating the implications between these concepts.


Typos

1. Line 13 Column 2: Li ⊂ U should have ⊆? And in other pleases also, e.g., in related work
2. In Line 66 Column 1, the citations should be either chronological or alphabetical? Currently, they seem to follow neither ordering.
3. Line 81 column 2 typo repeated “in terms”
4. Line 155 column 2, has a typo, one needs to consider the size of the set (to be less than ∞). The same issue is also present in other places, e.g., Definitions 2.4 and 2.5.
5. The notation “abbreviate a finite sequence x1, . . . , xn as x1:n.”  from line 183 should appear earlier in the preliminaries – before its first use.
6. Line 324 column 1, Line 347 column 1 etc. have some typos – which seem like a missing latex definition.
7. Some citations have “et al.” please present full citations

**Other Strengths And Weaknesses:**

This paper is very relevant to the ICML community and one of its strengths is that it tackles a fundamental problem in the foundations of generative AI; using a recent model of language generation.

**Questions For Authors:**

Q1. Could you please confirm if the issue I mentioned in the proof of Theorem 3.1 is correct, and could you check whether it could be fixed?

**Relation To Broader Scientific Literature:**

**Novelty compared to existing works**
As the authors agree, the results and techniques of this paper are similar to those of Li et al. For example, the definition of the noisy closure dimension is clean but is a simple (but not immediately obvious) extension of the closure dimension introduced by Li et al. While the definitions and proof structures are similar, I think there are enough differences to justify accepting this paper to ICML. Further, the paper studies a foundational problem that is fundamentally different from the problem studied in Li et al. The fact that the characterizations end up being similar in spirit as those of Li et al. should not be considered a weakness of this paper but rather a strength.

**Comparison to Related Work**
I think the paper does a good job of discussing recent related works and mentioning the related works on language identification in the noisy setting. To further improve the writing, it would be useful to compare the results in this paper to existing results on language identification with noise from earlier studies. Given space constraints, this discussion could be in an appendix; but it would be very useful for the reader to see a summary of the results from prior work

**Theoretical Claims:**

I did not carefully check the proofs of the theorems but I did sanity check that the results sound plausible based on the existing results on language generation.

---

> ### Author Rebuttal · Authors · 2025-03-26
>
> We thank the reviewer for their comments. We will make sure to address all typos (i.e., typos 1-7) and suggestions (i.e., suggestions 1-5) in the camera-ready version.
>
> > Could you please confirm if the issue I mentioned in the proof of Theorem 3.1 is correct, and could you check whether it could be fixed?
>
> We thank the reviewer for catching this bug! We agree with the reviewer's finding that our proof of Theorem 3.1 is incomplete since there exists classes $F$ such that $|\bigcap_{h \in F} \operatorname{supp}(h)| < \infty$, but for every $f \in F$, we still have that  $|\bigcap_{h \in F\setminus \{f\}} \operatorname{supp}(h)| < \infty$ (e.g. the example provided by the reviewer). However, the reviewer's fix to the proof of the necessity direction does *not* work, and the class they provided shows that the condition is Theorem 3.1 is *not* necessary.
>
> The reason is subtle and due to the fact that in Definition 2.4 (line 186-187), we decided to measure the "sample complexity" of the generator in terms of the number *positive* examples it has seen and not the *overall* number of unique examples, since the latter is too stringent (as evidenced by the reviewer's own proof). That is, as written in Definition 2.4 , the generator only needs to be correct after observing $d$ *positive* examples. This is unlike our definitions for noisy uniform, noisy non-uniform, and in-the-limit, where the "sample complexity" is defined with respect to the *total number* of examples (i.e the generator has to be correct after observing any $d$ examples). We agree this is confusing and we will change Definition 2.4 so that it matches the definitions for noisy uniform, noisy non-uniform, and in-the-limit. After doing so, the reviewer's necessity proof is now correct and the characterization in Theorem 3.1 holds for the modified version of uniform noisy generatability where we measure the sample complexity in terms of the *total number* of examples. We will make a note of this in the camera-ready version and acknowledge the reviewer for their contribution.
>
> Nevertheless, the current proof of the necessity direction of Theorem 3.1 gives the following necessary condition for Definition 2.4 as it is written: if there exists a subclass $F\subseteq  H$ and hypothesis $f \in F$ such that $|\bigcap_{h \in F} \operatorname{supp}(h)| < \infty$ and $|\bigcap_{h \in F\setminus \{f\}} \operatorname{supp}(h)| = \infty$, then $F$ is not uniformly noisily generatable when the sample complexity is measured in terms of positive examples. Note that all finite classes whose closure is finite satisfy this property. **Thus, the main takeaway of the section remains unchanged  -- uniform noisy generation is hard and only possible for finite classes that are generatable immediately.** We will correct Theorem 3.1 appropriately.
>
> The reviewer's counterexample highlights an interesting separation in generatability based on whether one measures the sample complexity in terms of only the positive examples or all examples. For completeness sake, we can "fix" Theorem 3.1  by providing a new characterization of uniformly noisy generatability when the sample complexity measures only the positive examples, like what is currently written in Definition 2.4.
>
> Claim: A class $H$ is uniformly noisily generatable if and only if $\sup_n (NC_n(H) - n) < \infty$.
>
> Proof sketch: For the necessity direction, suppose that $\sup_n (NC_n(H) - n) = \infty$. Then for every $d \in \mathbb{N}$, we can find a $t \geq d$ and a sequence $x_1, \dots, x_t$, such that $|\langle x_1, ..., x_t \rangle_{H, t-d}| < \infty.$ Hence, by padding $x_1, \dots, x_t$ with any remaining elements in $\langle x_1, ..., x_t \rangle_{H, t-d}$, we can force the Generator to make a mistake while ensuring that the hypothesis chosen is consistent with at least $d$ examples in the stream. For the sufficiency direction, if  $\sup_n (NC_n(H) - n) < \infty$, then there exists a $d \in \mathbb{N}$ such that for every $t \geq d$ and distinct $x_1, ..., x_t$, we have that either $\langle x_1, ..., x_t \rangle_{H, t-d} = \bot$ or $|\langle x_1, ..., x_t \rangle_{H, t-d}| = \infty.$ Thus, the algorithm which for $t \geq d$ plays from $\langle x_1, ..., x_t \rangle_{H, t-d}\setminus \lbrace{x_1, \dots, x_{t}\rbrace}$ if $\langle x_1, ..., x_t \rangle_{H, t-d} \neq \bot$ is guaranteed to succeed.
>
> To put this into context, if one measures sample complexity with respect to the *overall* number of examples, then another characterization of uniform noisy generatability is the finiteness of $\sup_n (NC_n(H))$. Thus, the difference between measuring the sample complexity with respect to just positive examples or with respect to all samples is whether or not you subtract $n$ from $NC_n(H)$.

---

### Official Review · Reviewer_Us4m · 2025-03-14

**Overall Recommendation:** 4

**Summary:**

The paper studies the problem of language generation in the limit in a noisy setting where an adversary inserts a finite number of negative examples. The paper provides necessary and sufficient conditions for when a binary hypothesis class can be noisily generatable. The paper examines various definitions of noisy generatability.

**Claims And Evidence:**

Yes

**Essential References Not Discussed:**

N/A

**Experimental Designs Or Analyses:**

N/A

**Methods And Evaluation Criteria:**

N/A

**Other Comments Or Suggestions:**

N/A

**Other Strengths And Weaknesses:**

1. The paper is well-written with nice intuition, and the math is rigorous.
2. The paper provides a complete characterization of which classes are noisily uniformly generatable.

**Questions For Authors:**

N/A

**Relation To Broader Scientific Literature:**

The paper extends the previous work by Kleinberg & Mullainathan (2024) to the noisy examples setting.

**Theoretical Claims:**

The proofs in the main text appear to be correct.

---

### Official Review · Reviewer_do42 · 2025-03-16

**Overall Recommendation:** 3

**Summary:**

The paper extends prior work on language generation by studying the generation of new, unseen positive examples even when the example stream is adversarially contaminated with a finite number of noisy (negative) examples.
It introduces new notions, namely noisy uniform generatability, noisy non-uniform generatability, and noisy generatability in the limit; these notions adapt the already introduced framework of uniform and non-uniform generation (as well as generation in the limit) to more realistic, noisy settings.
The authors also introduce the setting of uniform noisy generation, but prove that it is a significantly harder problem in that even hypothesis classes with just two hypotheses might not be uniformly noisily generatable.
A key contribution is the introduction of the Noisy Closure dimension, a scale-sensitive dimension that extends the Closure dimension (Li et al., 2024) and characterizes when a hypothesis class can be noisily generated, providing some sufficient and necessary conditions.
The authors further demonstrate that all countable classes are noisily non-uniformly generatable, ensuring they can eventually generate new positive examples even under finite noise in the stream.

**Claims And Evidence:**

The claims are clearly stated and supported by sufficient evidence.

**Essential References Not Discussed:**

The authors discussed the most relevant literature.

**Experimental Designs Or Analyses:**

N/A

**Methods And Evaluation Criteria:**

N/A

**Other Comments Or Suggestions:**

- The function $\\mathrm{supp}(\\cdot)$ is used but never formally defined. It would be better to have a definition, e.g., $\\mathrm{supp}(h) = h^{-1}(1)$.
- In the definition of $\\mathcal{H}(x\_{1:d},n)$ it would be better to use a semicolon (or a pipe) to separate examples from noise level, as it makes it easier to parse $\\mathcal{H}(x\_1, \\dots, x\_d; n)$ (see preliminaries and, e.g., line 345)
- Please, specify that you assume the natural numbers $\\mathbb{N}$ consist of the positive integers (excluding $0$), since it is not a universal assumption.
- At line 326, it could also be helpful to add the "mistake bound" from online learning as another helpful analogy other than the "sample complexity".
- Maybe specify that the assumption at line 356 in the proof of Lemma 3.8 is without loss of generality if that is the case.
- It seems that the inner loop of Algorithm 1 might always stop withing one of its iterations, and so the last line might be superfluous. For instance, in the proof of Theorem 3.9 at lines 385-392, the case $j = r\_s+1$ might contradict the properties of $\\mathcal{Q}$ since it is always given $x\_{r\_s}$ as input when generating $\\hat z\_1^s, \\dots, \\hat z\_{r\_s}^s$; hence, the if condition within the inner loops should be true for at least one $i \\in \[r\_s\]$.

Typos:
- Line 14: "by" instead of "by the".
- Line 62: "negative" instead of "positive".
- Line 91: "a countably" instead of "an countably".
- Line 122: "is possible" instead of "possible".
- Line 179: "countably infinite" instead of "countable infinite".
- Lines 182-183 should go at the beginning of Section 2, since the notation is already used before.
- Line 168: $\\in \\mathrm{supp}(h)$ instead of $\\subseteq \\mathrm{supp}(h)$, or put curly brackets around sequence.
- Line 202: "as a stream" instead of "of stream".
- Line 204: missing period at the end.
- Lines 190-197: $d^*$ instead of $d$ to be consistent with Definition 2.4.
- Line 228: "form" instead of "from".
- Lines 234-235: "any subset" instead of "the any subset", "for which there exists" instead of "there exists", $\\mathcal{F}\\setminus\\{f\\}$ instead of $\\mathcal{F}\\setminus f$.
- Line 245: $\\mathcal{F}\\setminus\\{f\\}$ instead of $\\mathcal{F}\\setminus f$.
- Line 287: "noise level" instead of "level" for clarity.
- Line 288: "exist" instead of "exists".
- Line 294: "as it is defined" instead of "it is defined".
- Line 312: $h \\in \\mathcal{H}$ instead of $h \\in \\mathrm{supp}(h)$.
- Line 319: "being" instead of "bring".
- Line 291: specify "positive examples".
- Line 295: "a subset of $\\mathcal{H}$" should refer more precisely to $\\mathcal{F}$.
- Line 338: specify Corollary 3.4 other than Theorem 3.3 for the result to follow.
- Line 381: "countable" instead of "countably infinite".
- Line 420: "at most" instead of "most".
- Lines 425-439, left column: all references to $t$ should actually be $s$, given line 423.
- Lines 385-392, right column: all references to $t$ should actually be $s$, given line 423.
- Line 397: "another" instead of "a another".

**Other Strengths And Weaknesses:**

A formal framework for the study of (language) generation was already introduced, as outlined by the authors themselves.
Nevertheless, handling the presence of noise in the stream is quite interesting and requires an adaptation of ideas from previous work, without being a direct consequence of them.
The authors also cover multiple settings where the difference lies in the dependence of the number of mistakes on the noise level and the hypothesis used to generate the positive examples, and extensively study their differences.

**Questions For Authors:**

- Do you believe it is possible to extend this generation framework beyond the "binary classification" one? For instance, consider generating a sequence with examples belonging to multiple classes and guaranteeing the generation of examples from some of these classes.
- Do you think the framework could be extended further, other than the multiclass one mentioned above? Do you foresee any technical challenge that would require significant changes in the framework or in the assumptions made?

**Relation To Broader Scientific Literature:**

I believe the main contributions of this work nicely extend the already available results on models for (language) generation to a more realistic setting with adversarial noise in the example stream.
As already remarkes, the authors do a good job in comparing to existing results and clearly framing their contribution relative to them.

**Theoretical Claims:**

Overall, the proofs of the claims are mainly correct.
Some parts, however, contain some mistakes that would need some adjustment.
More precisely, there is a wrong choice of indices in the proof of Theorem 3.9, as it seems that authors mistakenly swapped $s$ with $t$ starting from line 425.
One would then need to fix this and verify whether the following claims still hold true.
Additionally, in the proof of Theorem 3.1, the authors claim that $x\_1, \\dots, x\_d, z\_1, \\dots, z\_d$ contain $2d$ unique examples (lines 248 and 269) while this is not necessarily true, i.e., there can exist $i,j \\in \[d\]$ such that $x\_i = z\_j$.
This would require a different selection of those $2d$ examples in order to ensure their uniqueness.
Anyways, these are not major issues and they appear to be fixable at first glance.

---

> ### Author Rebuttal · Authors · 2025-03-26
>
> We thank the reviewer for their comments and suggestions. We agree with all the suggestions made by the reviewer and will make sure to incorporate these along with fixing the typos in the camera-ready version. Below, we address some questions and concerns.
>
> > More precisely, there is a wrong choice of indices in the proof of Theorem 3.9, as it seems that authors mistakenly swapped  with  starting from line 425. One would then need to fix this and verify whether the following claims still hold true.
>
> We thank the reviewer for catching this typo! The reviewer is exactly correct, and the claim goes through with this modification. We will make sure to make this change in the camera-ready version.
>
> > Additionally, in the proof of Theorem 3.1, the authors claim that...
>
> We thank the reviewer for this comment. However, we do believe that $x_1, \dots, x_d, z_1, \dots, z_d$ contains $2d$ points. To see why, suppose that $x_1, \dots, x_d$ is any set of $d$ distinct points in $\bigcap_{h \in F\setminus \{f\}} \operatorname{supp}(h).$ It suffices to show that $\{x_1, \dots, x_d\}$ and $\operatorname{supp}(f) \setminus \bigcap_{h \in F} \operatorname{supp}(h)$ are disjoint. Pick some $x_i \in \lbrace\{x_1, \dots, x_d\rbrace\}$. If $x_i \in \operatorname{supp}(f)$, then $x_i \in \bigcap_{h \in F} \operatorname{supp}(h)$. To see why, recall that $x_i \in \bigcap_{h \in F\setminus \{f\}} \operatorname{supp}(h)$ which means that $x_i \in \operatorname{supp}(h)$ for all $h \in  F\setminus \{f\}$. Thus, if $x_i $ is also in $\operatorname{supp}(f)$, then it must be the case that $x_i \in \bigcap_{h \in F} \operatorname{supp}(h)$. Overall, this means that $x_i \notin \operatorname{supp}(f) \setminus \bigcap_{h \in F} \operatorname{supp}(h).$ On the other hand, if $x_i \notin \operatorname{supp}(f)$, then it must be the case that $x_i \notin \operatorname{supp}(f) \setminus \bigcap_{h \in F} \operatorname{supp}(h),$ completing the proof.  We will make sure to clarify this in the camera ready version.
>
> > Do you believe it is possible to extend this generation framework beyond the "binary classification" one?
>
> Yes, we do believe that these results should generalize to the multiclass case through the similar setup of prompted generation from Li et al. (2024). For finite label spaces, the characterization of noisy generatability should remain unchanged and simply requires modifying the noisy closure dimension into the prompted noisy closure dimension. However, the more interesting case of infinite labels is unclear. We think this is an interesting direction of future work!
>
> > Do you think the framework could be extended further, other than the multiclass one mentioned above? Do you foresee any technical challenge that would require significant changes in the framework or in the assumptions made?
>
> Yes, we believe there are several frameworks for modeling noise that are of interest, beyond the multi class case and the setting we considered. For example,  one could also consider noisy generation in a stochastic setting like that studied by Kalavasis et a. [2024]. Moreover, one can also study variants of the noisy model where, for example, the injection of noise must follow a particular rate or where the generator has query access whether an example is noisy or not. In general, in the real-world, data is most likely not worst-case, and additional feedback is often available to the generator. Thus, it is an interesting direction to study relaxations of our model by either weakening the adversary or strengthening the generator. New techniques will need to be developed to understand how much such relaxations can help.
>
> Kalavasis, Alkis, Anay Mehrotra, and Grigoris Velegkas. "On the limits of language generation: Trade-offs between hallucination and mode collapse." arXiv preprint arXiv:2411.09642 (2024).

---

### Official Review · Reviewer_uTiP · 2025-03-17

**Overall Recommendation:** 4

**Summary:**

This paper proposes an extension of the theoretical model of language generation introduced by Kleinberg & Mullainathan (2024), exploring the impact of noisy example streams on generatability. The authors introduce the concepts of "noisy uniform generatability," "noisy non-uniform generatability," and "noisy generatability in the limit" to account for the influence of noise in the example stream, addressing a significant gap in previous research which assumed a noiseless setting. Key contributions include a complete characterization of noisy uniform generatability through the Noisy Closure dimension, new conditions for noisy non-uniform generatability, and sufficient conditions for noisy generatability in the limit. The paper also demonstrates that while noisy uniform generation is more difficult than its noiseless counterpart, all finite classes are still noisily uniformly generatable.

**Claims And Evidence:**

1. The paper extends Kleinberg & Mullainathan’s (2024) theoretical framework to include the impact of noisy example streams on generatability. Compared to the conclusions in noiseless settings, the new findings under noisy conditions are indeed a significant contribution. However, what are the technical difficulties and contributions, especially given that noise has already been extensively studied in non-generative contexts?
2. While this paper is theoretical and does not require experimental validation, would it be possible to conduct experiments? If so, how could one design and organize such experiments?
3. There appears to be a typo on line 63—should "positive examples" be "negative examples"?

**Essential References Not Discussed:**

N/A

**Experimental Designs Or Analyses:**

See the second point in Claims And Evidence

**Methods And Evaluation Criteria:**

The exploration of conclusions under noisy conditions is relevant and meaningful.

**Other Comments Or Suggestions:**

N/A

**Other Strengths And Weaknesses:**

N/A

**Questions For Authors:**

N/A

**Relation To Broader Scientific Literature:**

N/A

**Theoretical Claims:**

The theory appears to be sound.

---

> ### Author Rebuttal · Authors · 2025-03-26
>
> We thank the reviewer for their comments. We address their questions and concerns below.
>
> > However, what are the technical difficulties and contributions, especially given that noise has already been extensively studied in non-generative contexts?
>
> As the reviewer noted, noise has been extensively studied in non-generative contexts, for example in PAC and online learnability. However, to the best of our knowledge, this is the first work to rigorously formalize and study noise in the framework posed by Kleinberg \& Mullainathan (2024) and Li. et al. (2024). Our main contributions are summarized near the end of Section 1. At a high level:
>
> 1) We introduce a learning-theoretic theoretical framework for studying noise in language generation,
> 2) We provide necessary and sufficient conditions in terms of combinatorial dimensions for various notions of noisy generation, spanning different levels of difficulty for the generator.
>
> We feel that this work is an important stepping stone towards bridging learning theory and robustness in generative machine learning. As for technical difficulties, accounting for noise in the data stream requires new proof techniques. Indeed, since, in our model of noise, the generator does not know the number or location of noisy examples, it is a priori unclear whether and how generation of clean positive examples is possible. Nevertheless, it is possible, and our results show that indeed we need to go beyond existing results by considering scale-sensitive dimensions. This is unlike in prediction, where one does not need a scale-sensitive dimension to characterize agnostic PAC and online learnability.
>
> > While this paper is theoretical and does not require experimental validation, would it be possible to conduct experiments? If so, how could one design and organize such experiments?
>
> This is a great question, and one that we have been thinking about too! One high-level takeaway from our results is that in order to be robust to noisy examples in the training data, one needs to explicitly incorporate the fact that noisy data may be present in the training dataset during the process of training a generator. Moreover, our results hint at the fact that it might be useful to iteratively guess the level of noise in the training data and use this guess to adapt the training procedure. These two insights may lead to the following generation algorithm which could be interesting to study experimentally: start with an initial guess of the noise in the training set, use this initial guess to train a robust variant of a GAN that can take in a noise-level, use the robust GAN to estimate the amount of noise in the training data, and repeat this process.
>
> > There appears to be a typo on line 63—should "positive examples" be "negative examples"?
>
> The reviewer is exactly right! This should be negative examples. We will make sure to fix this in the final version.

---

### Decision · Program_Chairs · 2025-05-01

**Decision:**

Accept (poster)

**Comment:**

This work studies language generation (generation of new, unseen positive examples) in the presence of adversarial contamination with negative examples. A systematic investigation of this task is initiated and a characterization is given under some conditions. The reviewers agreed that this is an interesting contribution that merits acceptance.